# Eastern equine encephalitis virus rapidly infects and disseminates in the brain and spinal cord of cynomolgus macaques following aerosol challenge

Janice A. Williams[1], Simon Y. Long[1], Xiankun Zeng[1], Kathleen Kuehl[1], April M. Babka[1], Neil M. Davis[1], Jun Liu[1], John C. Trefry[2], Sharon Daye[1], Paul R. Facemire[1], Patrick L. Iversen[3], Sina Bavari[4], Margaret L. Pitt[2,4]*, Farooq Nasar[2]*

1 Pathology Division, United States Army Medical Research Institute of Infectious Diseases, Frederick, Maryland, United States of America, 2 Virology Division, United States Army Medical Research Institute of Infectious Diseases, Frederick, Maryland, United States of America, 3 Therapeutics Division, United States Army Medical Research Institute of Infectious Diseases, Frederick, Maryland, United States of America, 4 Office of the Commander, United States Army Medical Research Institute of Infectious Diseases, Frederick, Maryland, United States of America

☯ These authors contributed equally to this work.
* margaret.l.pitt.civ@mail.mil (MLP); farooq.nasar.ctr@mail.mil, fanasar@icloud.com (FN)

**Data Availability Statement:** All relevant data are within the manuscript and its Supporting Information files.

## Abstract

Eastern equine encephalitis virus (EEEV) is mosquito-borne virus that produces fatal encephalitis in humans. We recently conducted a first of its kind study to investigate EEEV clinical disease course following aerosol challenge in a cynomolgus macaque model utilizing the state-of-the-art telemetry to measure critical physiological parameters. Here, we report the results of a comprehensive pathology study of NHP tissues collected at euthanasia to gain insights into EEEV pathogenesis. Viral RNA and proteins as well as microscopic lesions were absent in the visceral organs. In contrast, viral RNA and proteins were readily detected throughout the brain including autonomic nervous system (ANS) control centers and spinal cord. However, despite presence of viral RNA and proteins, majority of the brain and spinal cord tissues exhibited minimal or no microscopic lesions. The virus tropism was restricted primarily to neurons, and virus particles (~61–68 nm) were present within axons of neurons and throughout the extracellular spaces. However, active virus replication was absent or minimal in majority of the brain and was limited to regions proximal to the olfactory tract. These data suggest that EEEV initially replicates in/near the olfactory bulb following aerosol challenge and is rapidly transported to distal regions of the brain by exploiting the neuronal axonal transport system to facilitate neuron-to-neuron spread. Once within the brain, the virus gains access to the ANS control centers likely leading to disruption and/or dysregulation of critical physiological parameters to produce severe disease. Moreover, the absence of microscopic lesions strongly suggests that the underlying mechanism of EEEV pathogenesis is due to neuronal dysfunction rather than neuronal death. This study is the first comprehensive investigation into EEEV pathology in a NHP model and will provide significant insights into the evaluation of countermeasure.

**Funding:** This study was supported by a grant from Medical Countermeasure Systems-Joint Vaccine Acquisition Program [Grant #A5XA0A7444182001 (FN and MLP)]. The funders had no role in study design, data collection, analysis, decision to publish, or preparation of the manuscript.

**Competing interests:** The authors have declared that no competing interests exist.

## Author summary

EEEV is an arbovirus endemic in parts of North America and is able to produce fatal encephalitis in humans and domesticated animals. Despite multiple human outbreaks during the last 80 years, there are still no therapeutic or vaccines to treat or prevent human disease. One critical obstacle in the development of effective countermeasure is the lack of insights into EEEV pathogenesis in a susceptible animal host. We recently conducted a study in cynomolgus macaques to investigate the disease course by measuring clinical parameters relevant to humans. Following infection, these parameters were rapidly and profoundly altered leading to severe disease. In this study, we examined the potential mechanisms that underlie pathogenesis to cause severe disease. The virus was present in many parts of the brain and spinal cord, however, minimal or no pathological lesions as well as active virus replication were observed. Additionally, neurons were the predominant target of EEEV infection and virus transport was facilitated via axonal transport system to spread neuron-to-neuron throughout the brain and spinal cord. These data show that EEEV likely hijacks essential transport system to rapidly spread in the brain and local/global neuronal dysfunction rather than neuronal death is the principal cause of severe disease.

## Introduction

The genus *Alphavirus* in the family *Togaviridae* is comprised of small, spherical, enveloped viruses with genomes consisting of a single stranded, positive-sense RNA, ~11–12 kb in length. Alphaviruses comprise 31 recognized species and the vast majority utilize mosquitoes as vectors for transmission into vertebrate hosts [1–6]. Mosquito-borne alphaviruses can spillover into the human population and cause severe disease. Old World alphaviruses (chikungunya, o'nyong-nyong, Sindbis, and Ross River) can cause disease characterized by rash and debilitating arthralgia, whereas New World viruses [eastern, western, and Venezuelan equine encephalitis virus] can cause fatal encephalitis.

Eastern equine encephalitis virus (EEEV) is an important pathogen of medical and veterinary importance in North America. EEEV is endemic in the eastern United States and Canada, and the Gulf coast of the United States. The main transmission cycle is between passerine birds and *Culiseta melanura* mosquitoes. However, this cycle can spillover into humans and domesticated animals and cause severe disease with human and equid case-fatality rates of 30–90% and >90%, respectively [6,7]. Human survivors can suffer from debilitating and permanent long-term neurological sequelae with rates of 35–80% [6,7]. In addition to natural infections, EEEV was developed as a biological weapon during the cold war by the U.S. and the former Union of Soviet Socialist Republics (USSR). Currently, there are no licensed therapeutics and/or vaccines to prevent or treat EEEV infection and the U.S. population remains vulnerable to natural disease outbreaks and/or bioterrorism events.

The development of effective vaccine and therapeutic countermeasures has utilized nonhuman primate (NHP) models to recapitulate various aspects of human disease, as well as, to gain insight into EEEV disease. We recently conducted a study in cynomolgus macaques to explore EEEV disease course utilizing advance telemetry to monitor critical physiological including temperature, respiration, activity, heart rate, blood pressure, electrocardiogram (ECG), and electroencephalography (EEG) following aerosol challenge at a dose of 7.0 $\log_{10}$ PFU/animal [8]. Following challenge all parameters were altered rapidly and considerably, and

accordingly, all NHPs met the euthanasia criteria by ~106–140 hours post-infection (hpi). Our previous report detailed the alterations of the parameters, however, the potential underlying mechanism/s responsible were not investigated [8]. Here, we report a comprehensive investigation into the pathology of NHP tissues collected at euthanasia to gain insights into EEEV pathogenesis.

## Materials and methods

### Ethics statement

This work was supported by an approved United States Army Medical Research Institute of Infectious Diseases (USAMRIID) Institutional Animal Care and Use Committee (IACUC) animal research protocol. Research was conducted under an IACUC approved protocol in compliance with the Animal Welfare Act, PHS Policy, and other Federal statutes and regulations relating to animals and experiments involving animals. The facility where this research was conducted is accredited by the Association for Assessment and Accreditation of Laboratory Animal Care, International and adheres to principles stated in the Guide for the Care and Use of Laboratory Animals, National Research Council, 2011 [9].

### Virus

Eastern equine encephalitis virus isolate V105-00210 was obtained from internal USAMRIID collection. The details of the stock have been described previously [8]. Briefly, the virus (Vero passage #1) was received from the Centers for Disease Control and Prevention (CDC) Fort Collins. The virus stock was passed in Vero-76 cells (American Type Culture Collection, ATCC; Bethesda, MD) twice to produce Master (Vero passage #2) and Working (Vero passage #3) virus stocks. The virus stock was deep sequenced to verify genomic sequence and to ensure purity. In addition, the stock was tested to exclude presence of endotoxin and mycoplasma.

### Non-human primate study design

A detailed study design was described previously [8]. Briefly, four (2 males, 2 females) cynomolgus macaques (*Macaca fascicularis*) of Chinese origin ages 5–8 years and weighing 3–9 kg were obtained (Covance). All NHPs were prescreened and determined to be negative for Herpes B virus, simian T-lymphotropic virus 1, simian immunodeficiency virus, simian retrovirus D 1/2/3, tuberculosis, *Salmonella* spp., *Campylobacter* spp., hypermucoviscous *Klebsiella* spp., and *Shigella* spp. NHPs were also screened for the presence of neutralizing antibodies to EEEV, VEEV IAB, and WEEV by plaque reduction neutralization test ($PRNT_{80}$). The NHPs were challenged with a target dose of $7.0 \log_{10}$ PFU of EEEV via the aerosol route utilizing in the head-only Automated Bioaerosol Exposure System (ABES-II). Following challenge, all four NHPs rapidly exhibited severe disease and met the euthanasia criteria ~106–140 hpi. Lung, liver, spleen, kidney, heart, spinal cord, and brain tissues were collected from each NHP at the time of euthanasia. Tissues were fixed for >21 days in 10% neutral buffered formalin, sectioned, and examined.

### Tissues processing and histopathology

Tissue sections from various organs were generated (S1 Table). NHP tissues were processed in a Tissue-Tek VIP-6 vacuum infiltration processor (Sakura Finetek USA, Torrance, CA) followed by paraffin embedding with a Tissue-Tek model TEC (Sakura). Sections were cut on a Leica model 2245 microtome at 4 μm, stained with hematoxylin and eosin (H&E) and coverslipped. Slides were examined by an ACVP diplomate veterinary pathologist blinded to

intervention. All images were captured with a Leica DM3000 microscope and DFC 500 digital camera using Leica Application Suite version 4.10.0 (Leica Microsystems, Buffalo Grove, IL).

### *In situ* hybridization

*In situ* hybridization (ISH) was performed using the RNAscope 2.5 HD RED kit (Advanced Cell Diagnostics, Newark, CA, USA) according to the manufacturer's instructions. Briefly, EEEV ISH probe targeting nucleotides 8680–9901 of EEEV isolate V105-00210 was designed and synthesized by Advanced Cell Diagnostics (Cat# 455721). Tissue sections were deparaffinized with Xyless II (Valtech, Brackenridge, PA, USA), followed by a series of ethanol washes and peroxidase blocking, then heated in kit-provided antigen retrieval buffer, and digested by kit-provided proteinase. Sections were exposed to ISH target probe pairs and incubated at 40˚C in a hybridization oven for 2 h. After rinsing with wash buffer, ISH signal was amplified using kit-provided Pre-amplifier and Amplifier conjugated to alkaline phosphatase and incubated with Fast Red substrate solution for 10 mins at room temperature. Sections were then stained with hematoxylin, air-dried, and mounted. ISH images were collected using an Olympus BX53 upright microscope (Olympus Scientific Solutions Americas Corp., Waltham, MA, USA).

### Immunohistochemistry

Immunohistochemistry (IHC) was performed using the Dako Envision system (Dako Agilent Pathology Solutions, Carpinteria, CA, USA). After deparaffinization and peroxidase blocking, sections were covered with Rabbit anti-alphavirus polyclonal antibody was obtained from internal USAMRIID stocks. The polyclonal antibody was generated as a reagent by vaccinating animals with E3-E2-6K-E1 antigen of EEEV. The antibody was used at a dilution of 1:5000 and incubated at room temperature for 30 minutes. They were rinsed, and treated sequentially by an HRP-conjugated, secondary anti-rabbit polymer (Cat. #K4003, Dako Agilent Pathology Solutions). All slides were exposed to brown chromogenic substrate DAB (Cat. #K3468, Dako Agilent Pathology Solutions), counterstained with hematoxylin, dehydrated, cleared, and coverslipped. IHC images were collected using an Olympus BX53 upright microscope (Olympus Scientific Solutions Americas Corp., Waltham, MA, USA).

### Immunofluorescence assay

Formalin-fixed paraffin embedded (FFPE) tissue sections were deparaffinized using xylene and a series of ethanol washes. After 0.1% Sudan black B (Sigma) treatment to eliminate the autofluorescence background, the sections were heated in Tris-EDTA buffer (10mM Tris Base, 1mM EDTA Solution, 0.05% Tween 20, pH 9.0) for 15 minutes to reverse formaldehyde cross-links. After rinses with PBS (pH 7.4), the sections were blocked with PBS containing 5% normal goat serum overnight at 4˚C. Then the sections were incubated with Rabbit anti-EEEV antibody (USAMRIID, 1:1000) and chicken anti-NeuN antibody (Abcam, 1:25), or chicken anti-GFAP (Abcam, 1:500, or mouse anti-CD68 (Agilent/Dako, 1:200) for 2 hours at room temperature. After rinses with PBS, the sections were incubated with secondary goat anti-chicken Alex Fluor 488 (green, 1:500), goat anti-rabbit Alex Flour 488 (green), goat anti-rabbit Cy3 (red), and/ or goat anti-mouse Cy3 antibodies (red, 1:500) for 1 hour at room temperature. Sections were cover slipped using the Vectashield mounting medium with DAPI (Vector Laboratories). Images were captured on an LSM 880 Confocal Microscope (Zeiss, Oberkochen, Germany) and processed using open-source ImageJ software (National Institutes of Health, Bethesda, MD, USA).

## Transmission electron microscopy

Formalin-fixed thalamic tissue from each NHP was obtained and submerged in 2.5% glutaraldehyde and 2% paraformaldehyde in 0.1M sodium phosphate buffer for further fixation. Samples were fixed for at least 24 hours at 4° C and then rinsed with milliQ-EM grade water, rinsed again with 0.1M sodium cacodylate buffer before post-fixing with 1% osmium tetroxide in 0.1M sodium cacodylate for 60 minutes. After osmium fixation, the samples were rinsed with 0.1M sodium cacodylate buffer, followed by a water wash then subjected to uranyl acetate *en bloc*. Samples were washed with water then dehydrated through a graded ethanol series including 3 exchanges with 100% ethanol. Samples were further dehydrated with equal volumes of 100% ethanol and propylene oxide followed by two changes of propylene oxide. Samples were initially infiltrated with equal volumes of propylene oxide and resin (Embed-812; EMS, Hatfield, PA, USA) then incubated overnight in propylene oxide and resin. The next day, the samples were infiltrated with 100% resin embedded and oriented in 100% resin and then allowed to polymerize for 48 hours at 60° C. 1 μm thick sections were cut from one tissue block and a region of interest for thin sectioning was chosen. 80nm thin sections were cut and collected on 200 mesh copper grids. Two grids from each sample were further contrast stained with 2% uranyl acetate and Reynold's lead citrate. Samples were then imaged on the Jeol 1011 TEM at various magnifications.

# Results

## EEEV associated pathology in the visceral organs

Visceral organs including the heart, liver, lung, kidney, and spleen were collected from all NHPs at the time of euthanasia and examined for virus and/or host induced pathology (S1 Table). There were no EEEV associated necrotic and/or inflammatory lesions in the visceral organs of any of NHPs (Fig 1 and S1 Table). In addition, *in situ* hybridization (ISH) and immunohistochemistry (IHC) were unable to detect presence of viral RNA or proteins in any organs, respectively (S1 and S2 Figs).

## EEEV associated pathology in the brain and spinal cord

The projections from the olfactory bulb connect to the amygdala and hippocampus via the primary olfactory cortex. We previously reported the presence of infectious virus in the olfactory bulb of NHPs at the time of euthanasia with titers ranging from 4.1–7.9 $\log_{10}$ PFU/g [8]. Accordingly, the amygdala and hippocampus were investigated for virus and/or host induced pathology. Mild to moderate necrotic and inflammatory lesions were observed throughout both regions of the brain in all NHPs (Fig 2). The necrotic lesions were characterized by neuronal degeneration, satellitosis, and necrosis, as well as vacuolation of the neuropil (Fig 2). The inflammatory lesions comprised predominantly of neutrophilic infiltrates in all NHP sections except the hippocampus of NHP #2. Lastly, viral RNA and proteins were readily detected throughout the amygdala and hippocampus of all NHPs (Fig 2).

The projection of the amygdala and hippocampus connect to the midbrain, which in turn connects to both the forebrain and the hindbrain. We next examined various structures in these regions including the hypothalamus, thalamus, corpus striatum, mesencephalon, medulla oblongata, frontal cortex, and cerebellum for virus and/or host induced pathology (S1 Table). In contrast to the pathology observed in the amygdala and hippocampus, the majority of these tissue sections displayed minimal or no microscopic lesions (Figs 3 and 4). Few focal lesions were observed in some regions and were restricted primarily to the corpus striatum,

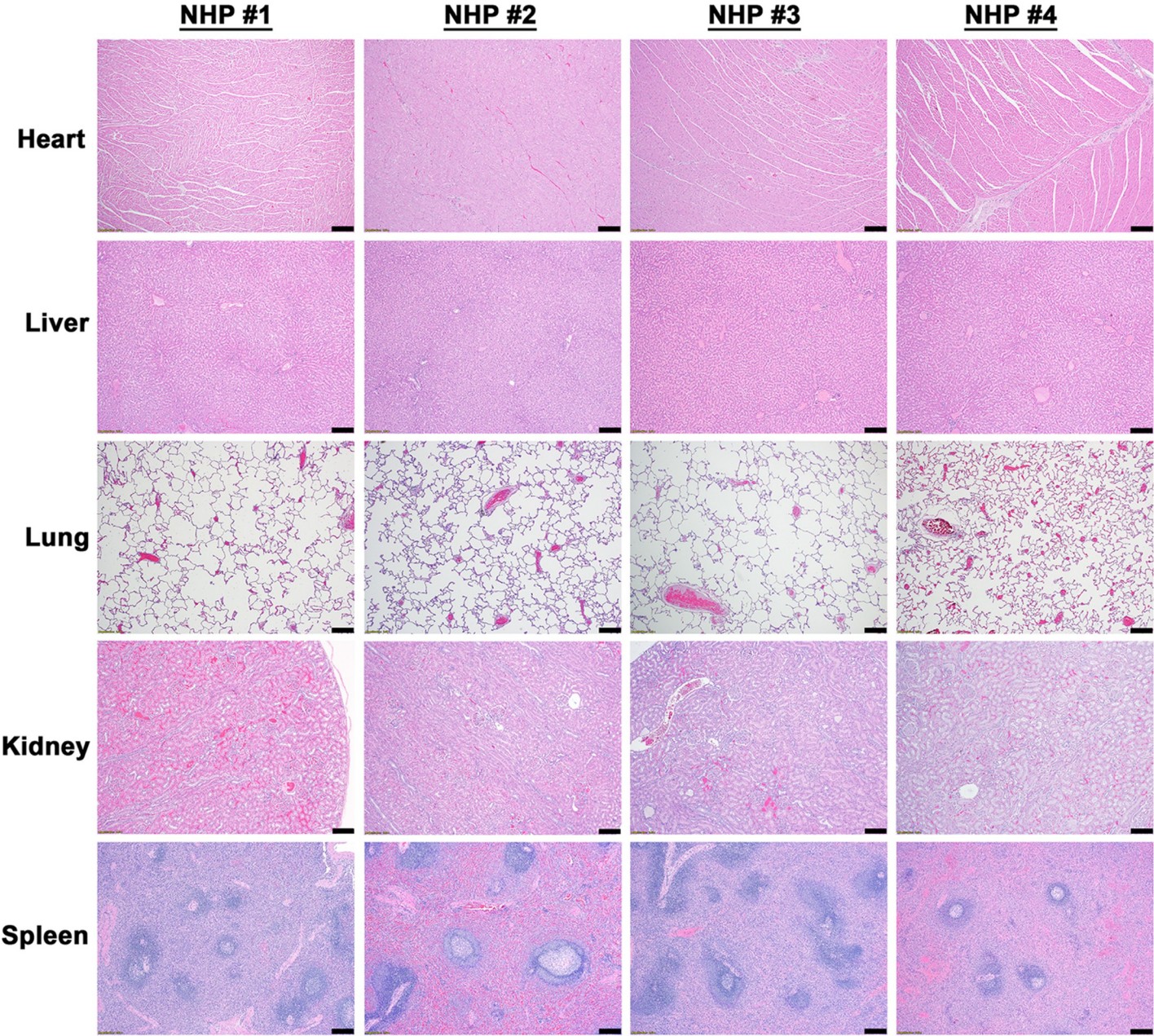

**Fig 1. Histopathology in the visceral organs of EEEV infected cynomolgus macaques.** The tissues were collected at the time of euthanasia. Hematoxylin and eosin (H&E) staining was performed on the tissues of all four NHPs. Bar = 200 um.

thalamus, mesencephalon, and medulla oblongata (Fig 4). The focal lesions comprised of minimal to mild neuronal degeneration, neuronal necrosis, neuropil vacuolation, gliosis, and neuronal satellitosis (Fig 4). The latter was most pronounced in the corpus striatum of NHP #1, which also displayed mild microhemorrhages (Fig 4). NHPs displayed mild to marked neutrophilic inflammation in the brain extending into the meninges. Additionally, perivascular infiltrates ranged from minimal lymphocytic and neutrophilic, to moderate and predominantly neutrophilic (Fig 4). The ISH staining detected substantial viral RNA in the brain tissue of all

# Amygdala

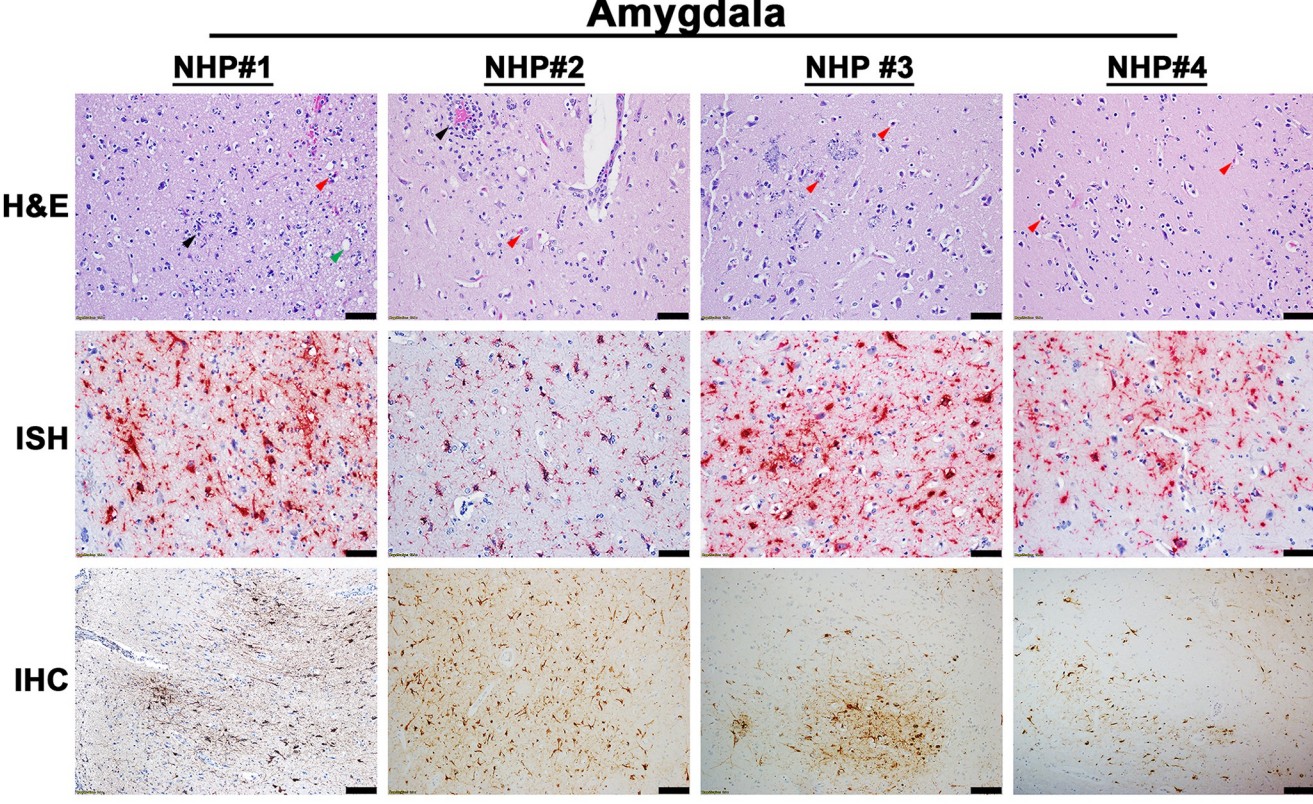

# Hippocampus

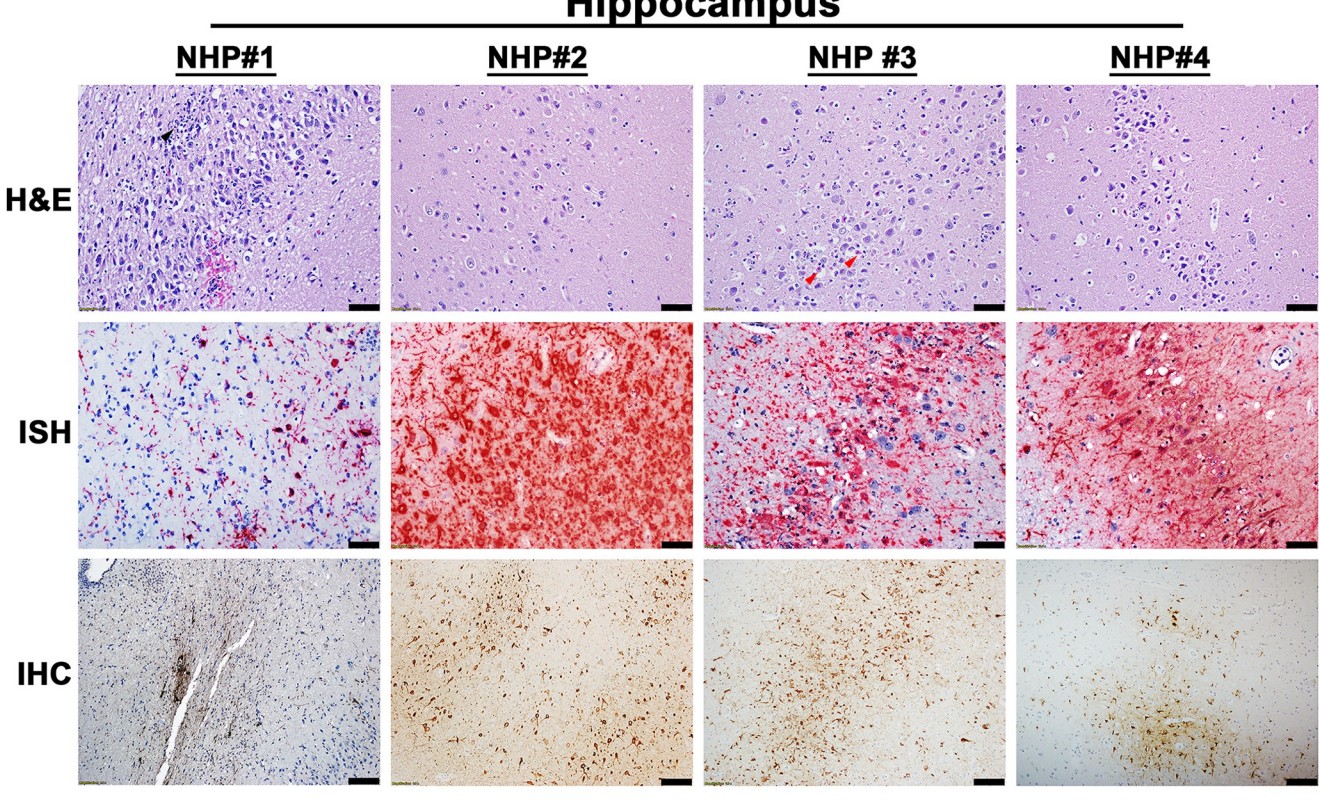

**Fig 2. Pathology in the amygdala and the hippocampus of EEEV infected cynomolgus macaques.** The tissues were collected at the time of euthanasia. Hematoxylin and eosin (H&E) staining was performed to visualize histopathology. The presence of EEEV RNA and proteins was visualized via *in situ* hybridization (ISH) and immunohistochemistry (IHC), respectively. H&E, ISH, and IHC were performed on the tissues of all four NHPs. Bar = 100 um (H&E and IHC). Bar = 50 um (ISH). Arrow key: neutrophilic infiltration/neutrophils (black), degenerative/necrotic neurons (red), and vacuolation of the neuropil (green).

NHPs (Fig 5). The IHC staining showed mild to marked immunoreactivity of neurons in all sections of the brain with the most pronounced in the corpus striatum, thalamus, mesencephalon, and medulla oblongata (Fig 6).

The cervical, thoracic, and lumbar spinal cord were also examined (S1 Table). In contrast to the brain sections, all three sections of the spinal cord displayed minimal or no pathological lesions (Fig 7). The main feature observed in the spinal cord sections was comprised of minimal inflammation. Viral RNA was readily detected in the cervical spinal cord of all four NHPs via ISH, whereas minimal or no RNA was detected in the thoracic and lumbar sections in three of the four NHPs (Fig 8). Substantial viral RNA was detected in thoracic and lumbar sections of NHP #3 (Fig 8). This finding was further verified by IHC staining that displayed a similar pattern (Fig 9).

## EEEV cell tropism in the thalamus of the NHPs

After establishment of EEEV infection in various brain regions, we next investigated virus tropism in the thalamus of all infected NHPs by determining infection in the astrocytes, microglia, and the neurons. Tissue sections were stained for viral RNA and cellular markers of astrocytes (GFAP), microglia (CD68), and neurons (NeuN) (Figs 10, 11, and 12). Minimal or no overlap was observed between viral RNA and GFAP or CD68, indicating minimal or no infection in the astrocytes and microglia, respectively (Figs 10 and 11). In contrast, considerable overlap of viral RNA and NeuN was observed in all NHPs suggesting that the majority of the viral infection was limited to the neurons (Fig 12).

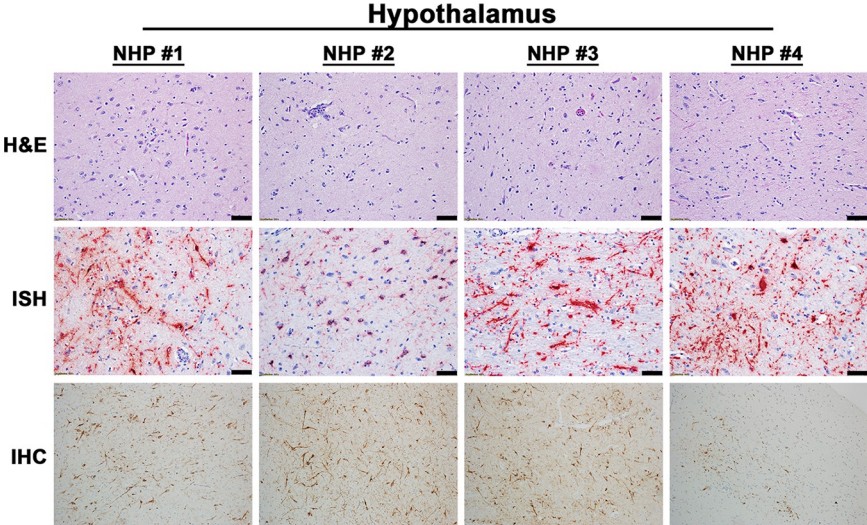

**Fig 3. Pathology in the hypothalamus of EEEV infected cynomolgus macaques.** The tissue was collected at the time of euthanasia. Hematoxylin and eosin (H&E) staining was performed to visualize histopathology. The presence of EEEV RNA and proteins was visualized via *in situ* hybridization (ISH) and immunohistochemistry (IHC), respectively. H&E, ISH, and IHC were performed on the tissues of all four NHPs. Bar = 100 um (H&E and IHC). Bar = 50 um (ISH).

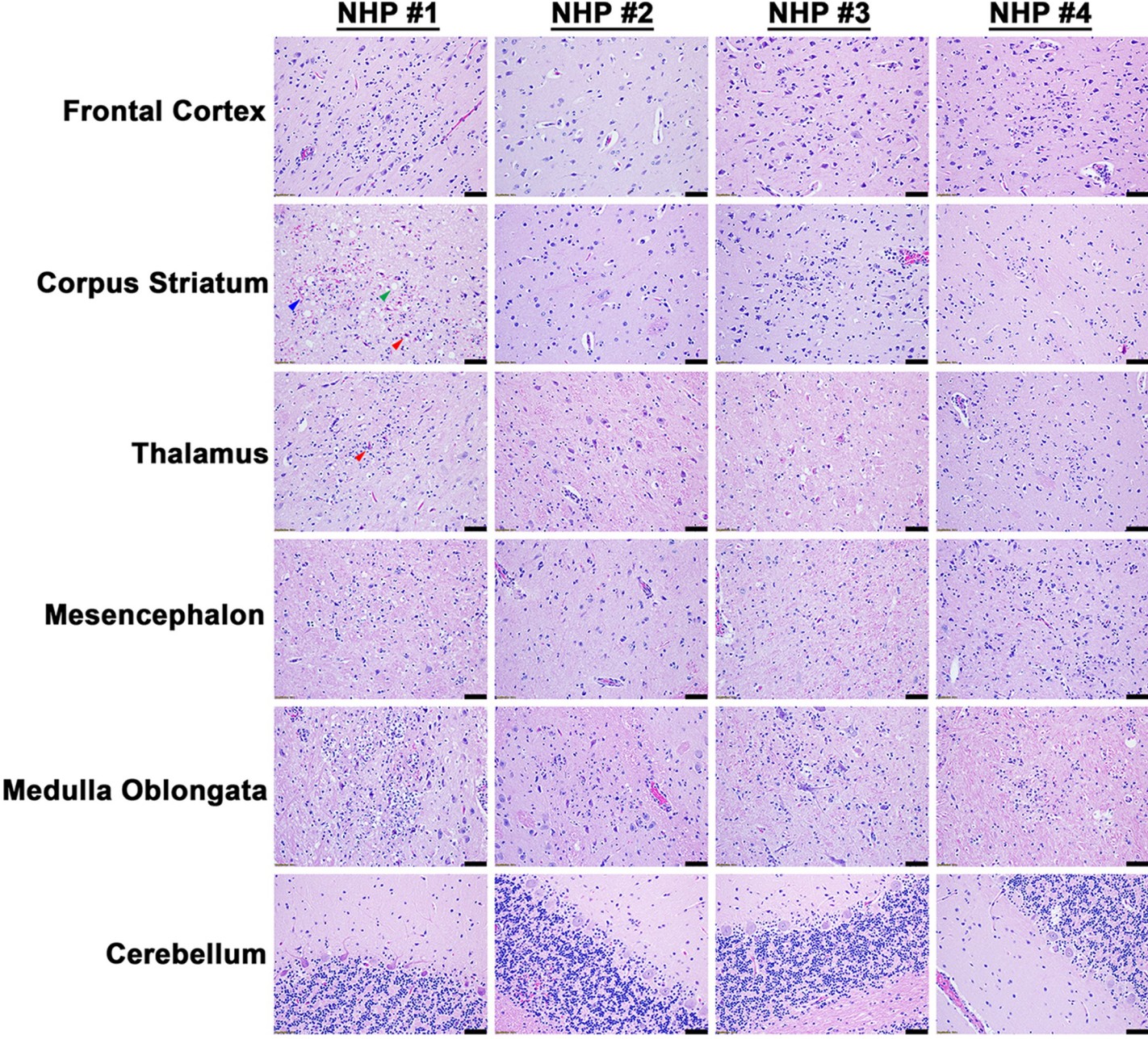

**Fig 4. Histopathology in various parts of the brain tissues of EEEV infected cynomolgus macaques.** The tissues were collected at the time of euthanasia. Hematoxylin and eosin (H&E) staining was performed to visualize histopathology. H&E was performed on the tissues of all four NHPs. Bar = 100 um. Arrow key: degenerative to necrotic neurons (red), vacuolation of the neuropil (green), and microhemorrhage (blue).

## Localization of EEEV virions in the thalamus of the NHPS via transmission electron microscopy (TEM)

The morphological analysis of various brain structures by the TEM showed no overt signs of apoptosis and/or necrosis as the majority of tissue sections displayed intact mitochondria and nuclei. TEM analysis showed marked presence of EEEV particles in the extracellular spaces throughout the thalamus of all NHPs (Figs 13 and S3). The majority of the virus particles were

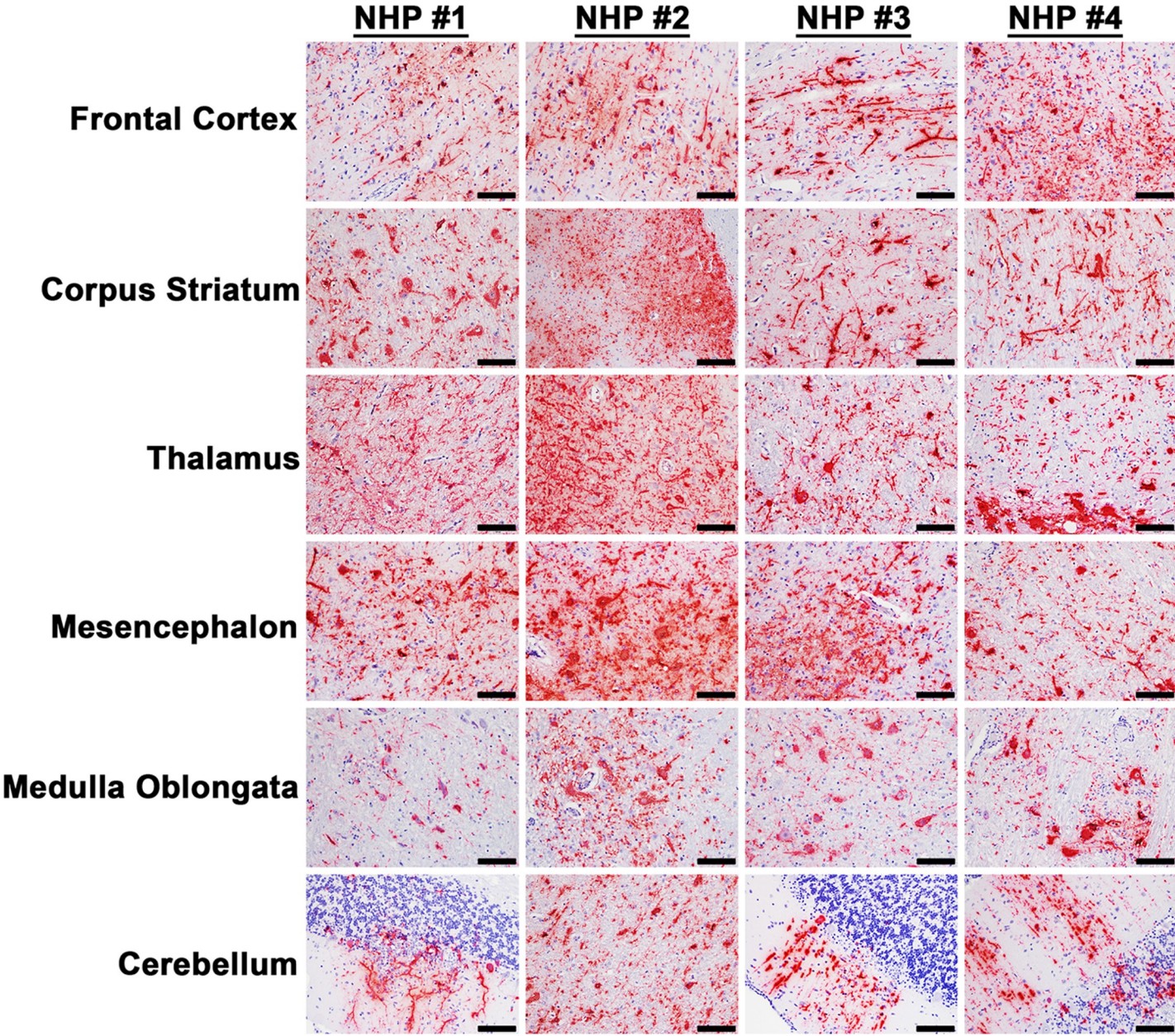

**Fig 5. The presence of EEEV RNA in various parts of the brain tissues of infected cynomolgus macaques.** The tissues were collected at the time of euthanasia. The presence of viral RNA was visualized via *in situ* hybridization (ISH). ISH was performed on the tissues of all four NHPs. Bar = 50 um.

spherical, ~61–68 nm in diameter, and were in close proximity to plasma membranes of the surrounding cells (Fig 14). Virus particles were detected juxtaposed to myelin sheaths, surrounding the axons, as well as near synapses (Figs 15 and S4).

The intracellular localization of EEEV within the thalamus was examined by detecting the presence of EEEV particles within the axons of neurons. Virus particles, ~62–67 nm in diameter, were detected within the axons in all NHPs (Fig 16). Surprisingly, the majority of the particles were not contained within vesicles and appeared to be free virions. In two sequential sections, ~80 nm apart, of an axon, the quantity of EEEV virions present inside an axon was assessed. The sections showed the presence of 18 and 17 particles, respectively (Fig 17A and 17B). This finding

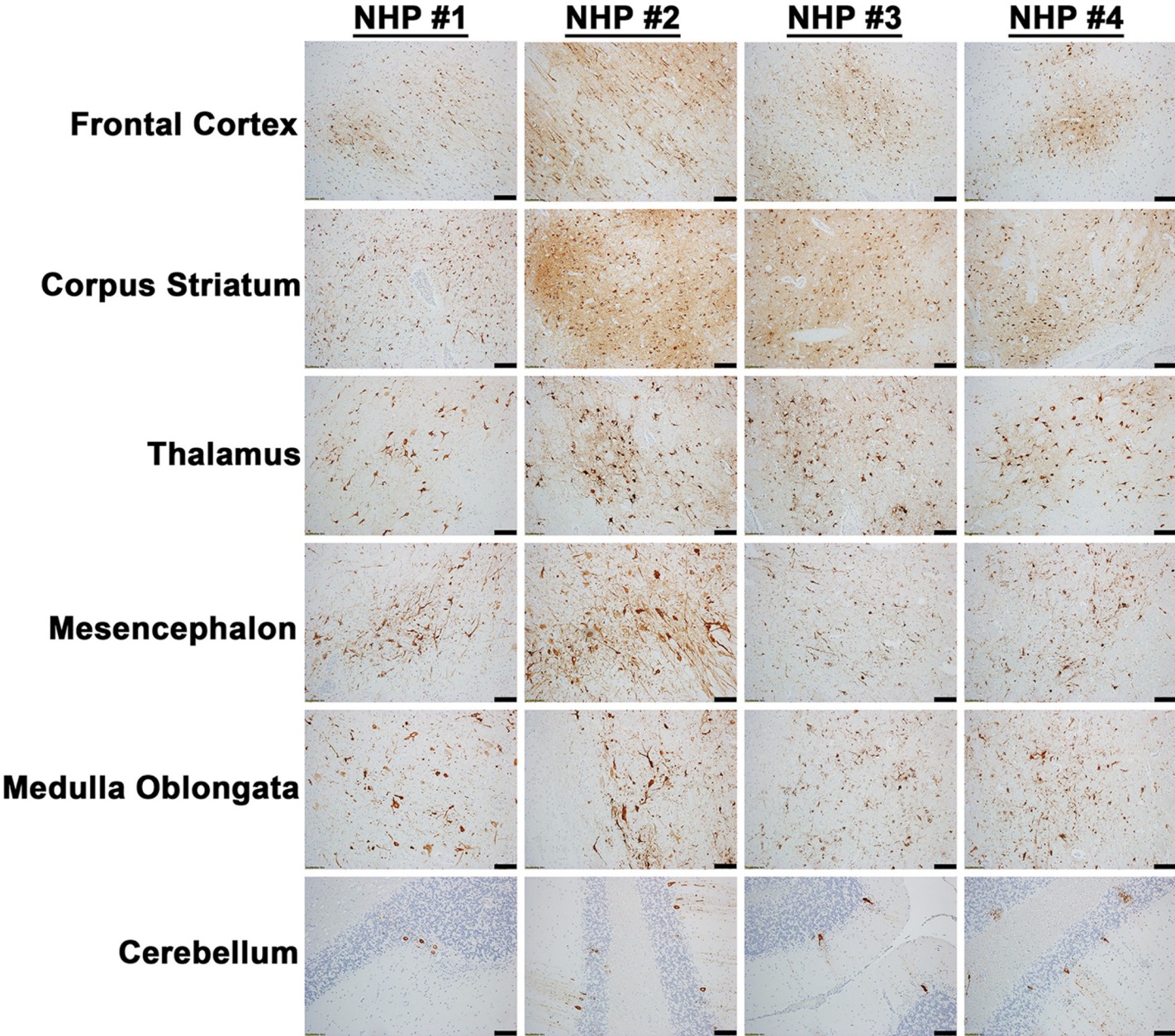

**Fig 6. The presence of EEEV proteins in various parts of the brain tissues of infected cynomolgus macaques.** The tissues were collected at the time of euthanasia. The presence of viral proteins was visualized via immunohistochemistry (IHC). IHC was performed on the tissues of all four NHPs. Bar = 100 um.

highlights the potential of large quantity of particles that can migrate through a single axon to infect other neurons.

Next, we sought to investigate active virus replication centers by detecting cytopathic vacuoles and budding virions in infected cells. Cells with active replication were rare in the brain, however, they were detected in all NHPs. The majority of the virus replication was localized to the amygdala, hippocampus, thalamus, and hypothalamus (Figs 18, 19, and S5). Extensive cytopathic vacuoles with attached and free nucleocapsid, ~40 nm in diameter, were present within the cytoplasm of infected cells (Figs 18 and 19). Furthermore, infectious virions ~65–68 nm in diameter were observed budding from infected host cell plasma membrane (Fig 19).

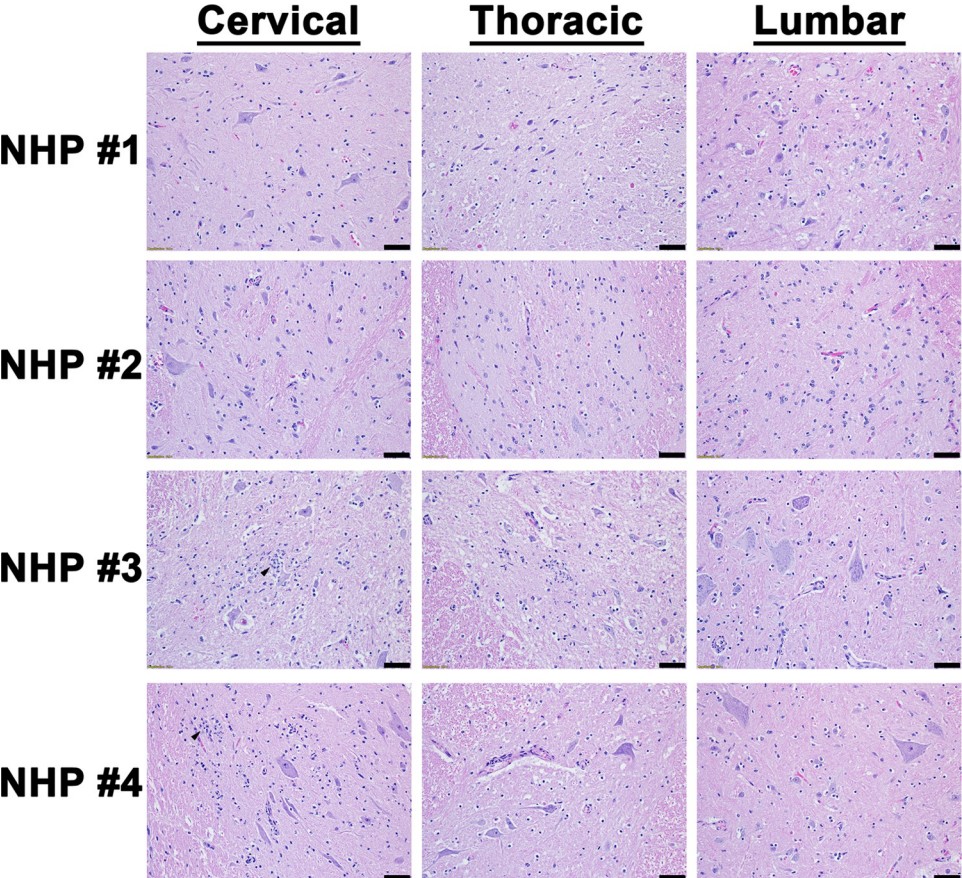

**Fig 7. Histopathology in various parts of the spinal cord of EEEV infected cynomolgus macaques.** The tissues were collected at the time of euthanasia. Hematoxylin and eosin (H&E) staining was performed to visualize histopathology. H&E was performed on the tissues of all four NHPs. Bar = 100 um. Arrow key: neutrophilic infiltration/neutrophils (black).

Another interesting finding in the TEM experiments was the presence of virus particles enclosed within undefined vesicular compartments in the extracellular space of all NHP tissues (S6 Fig). The vesicles were composed either exclusively of virions or mixture of particles and cellular component of similar size and shape (S6 Fig). Free virions could also be found adjacent to the enclosed vesicle (S6A and S6C Fig).

Although rare, necrotic lesions were visible within the thalamus and were detected by TEM. Considerable degeneration of the cellular architecture was observed with loss membrane integrity, disintegration of organelles, and cell lysis (S7 Fig). EEEV particles were readily detected scattered throughout the remaining cell debris (S7 Fig).

## Discussion

The susceptibility of cynomolgus macaques to EEEV via the aerosol route has been explored previously, however, pathology was not the main focus thus the data are limited [10–13]. The principal findings in the CNS and the spinal cord were comprised of meningoencephalitis and vasculitis with histological features including neuronal necrosis, perivascular cuffs, cellular debris, gliosis, satellitosis, edema, hemorrhage, with neutrophil and lymphocytic infiltrates. Our data are in agreement with the majority of these previous findings. The brain regions

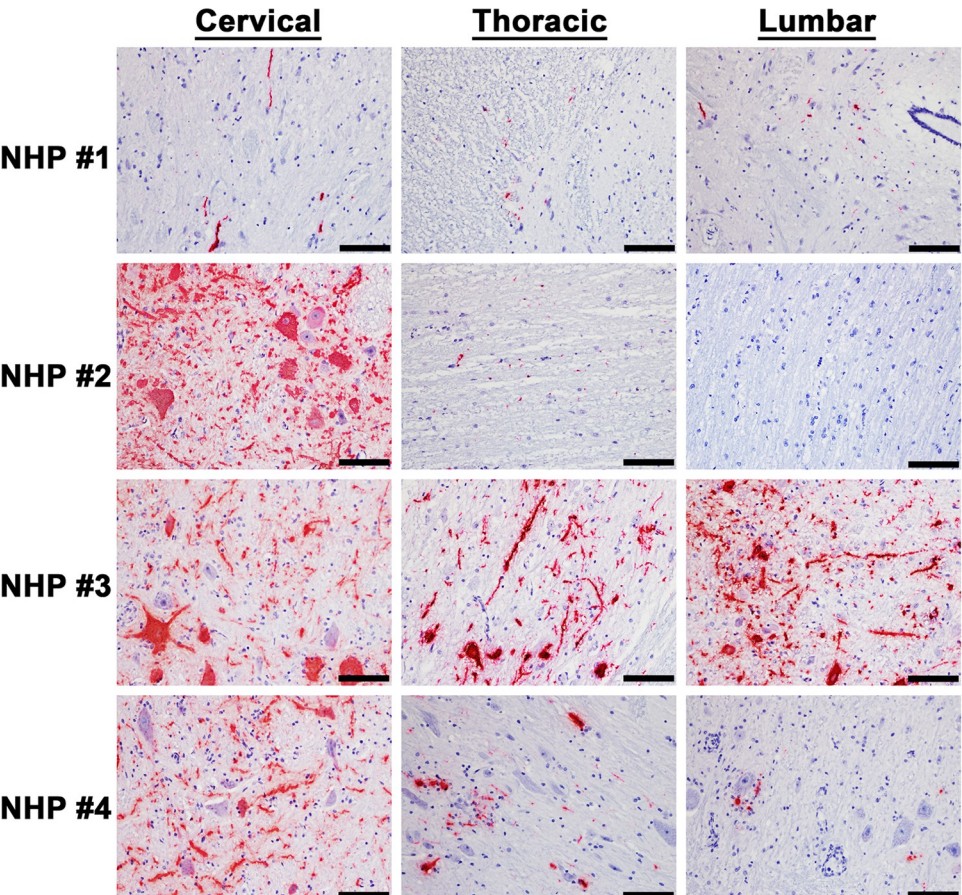

**Fig 8. The presence of EEEV RNA in various parts of the spinal cord of infected cynomolgus macaques.** The tissues were collected at the time of euthanasia. The presence of viral RNA was visualized via *in situ* hybridization (ISH). ISH was performed on the tissues of all four NHPs. Bar = 50 um.

proximal to the olfactory bulb such as hippocampus and medulla displayed meningoencephalitis and similar histological features. However, in our study the distal regions of the brain and spinal cord exhibited minimal or no microscopic lesions despite the presence of viral RNA and proteins. The potential explanation of the difference may be possible attenuation of EEEV isolates utilized in previous studies due to different methods of propagation to produce viral stocks. The average dose in our study was 6.6 log$_{10}$ PFU/NHP and was uniformly lethal by 4–6 dpi [8]

The data from this study provide insights into the mode of virus dissemination following aerosol challenge. Post-challenge, there are two potential routes for virus dissemination in the NHP host. The initial virus replication in the respiratory tract followed by systemic infection and subsequent access to central nervous system (CNS). Alternatively, the olfactory epithelium and bulb could serve as the initial site of virus replication followed by virus transport and infection of the olfactory tract with spread to the distal regions of the brain. The data from our study showed no evidence of gross and/or microscopic changes, as well as viral RNA or proteins in the lung, liver, heart, spleen, and kidneys. In contrast, pathological lesions were detected in the brain comprising of neutrophilic inflammation, neuronal degeneration, and necrosis. EEEV RNA and proteins were also readily detected throughout many parts of the brain and spinal cord. A gradation of viral RNA and proteins was observed the most in the

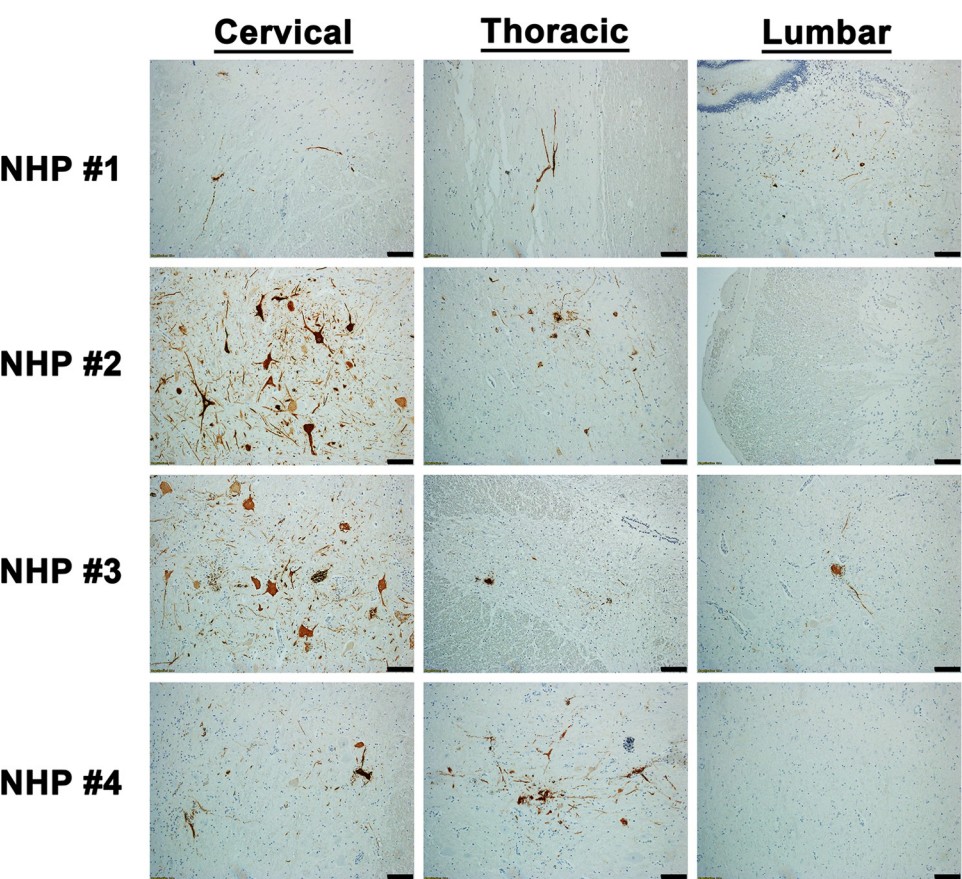

**Fig 9. The presence of EEEV proteins in various parts of the spinal cord of infected cynomolgus macaques.** The tissues were collected at the time of euthanasia. The presence of viral proteins was visualized via immunohistochemistry (IHC). IHC was performed on the tissues of all four NHPs. Bar = 100 um.

cervical region due to its proximity to the brainstem and least in the lumbar region. Lastly, we previously reported the presence of high EEEV infectious titers at the olfactory bulb of all NHPs [8]. Taken together, these data support the rapid and direct spread of EEEV via the olfactory bulb into the brain followed by dissemination into the spinal cord.

Although limited pathology was investigated in previous macaque studies, the dissemination of EEEV following aerosol infection was not investigated; however, it has been examined in mice, guinea pigs, and marmosets [14–17]. In these previous studies, virus was localized almost exclusively in the brain and was readily detected in the frontal cortex, corpus striatum, thalamus, hippocampus, mesencephalon, pons, medulla oblongata, and cerebellum [14–17]. In contrast, EEEV could not be detected in the heart, liver, lung, spleen, and kidney of guinea pigs and marmosets [14,17]. Murine studies displayed similar pattern to guinea pigs and marmosets, however, EEEV was detected lung and heart [15,16]. Our NHP data are in agreement with the guinea pig and marmoset studies. The presence of virus in the mouse lung and heart tissues shows important differences between the murine and other animal models.

In nature, EEEV is transmitted via a mosquito bite and can cause fatal encephalitis in many mammalian species including horses, sheep, cattle, alpacas, llamas, deer, dogs, pigs, and humans [18–60]. Following the bite of an infected mosquito, the virus replicates locally in skeletal muscle cells, fibroblasts, and osteoblasts, gains access to the peripheral tissues and organs, and eventually disseminates into the CNS to cause fatal encephalitis [61]. During the course of

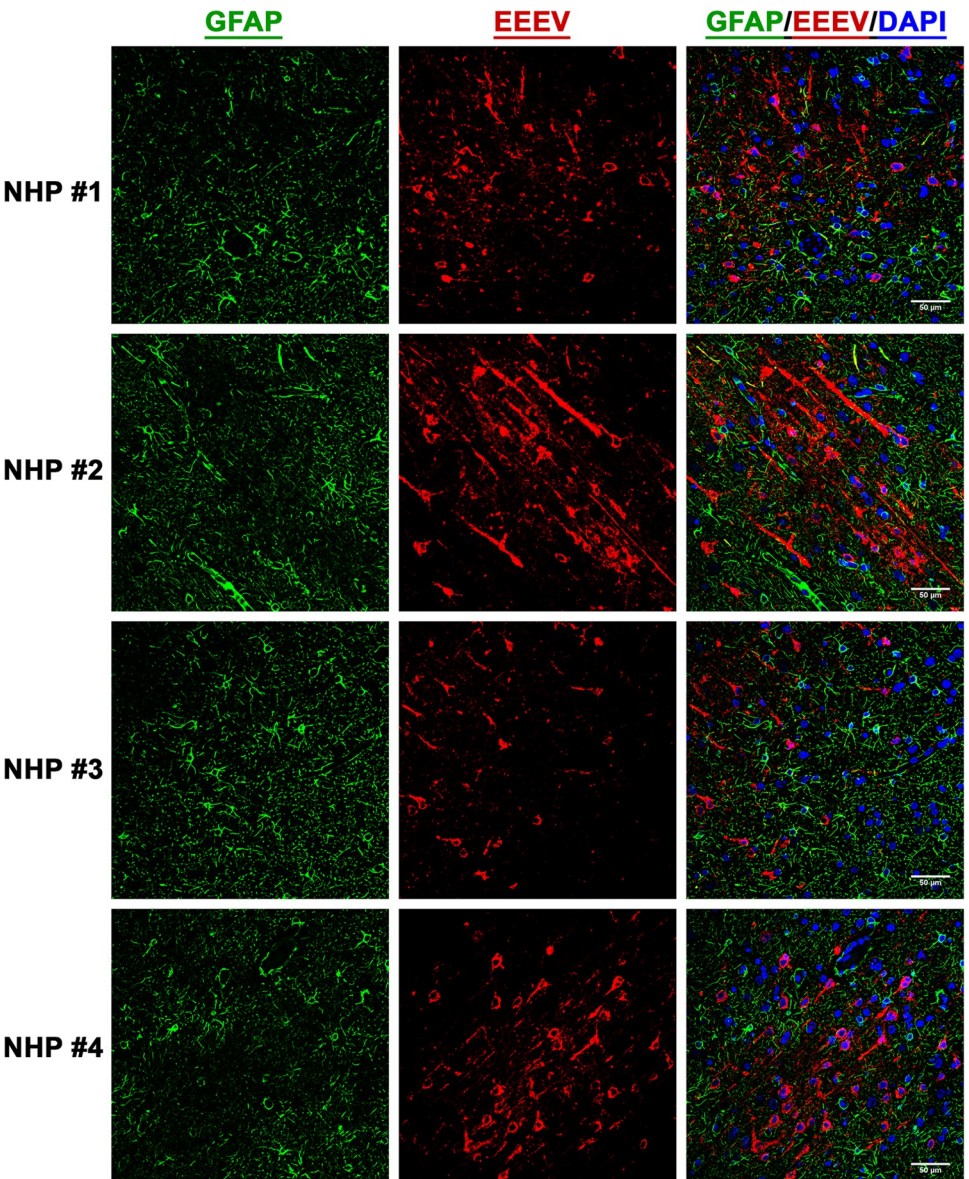

**Fig 10. The presence of EEEV RNA in the astrocytes of infected cynomolgus macaques.** Sections from the thalamus of each NHP were visualized via immunofluorescence assay. Sections were stained for GFAP (green), EEEV (red), and DAPI (blue).

infection, extensive pathology is observed in the visceral tissues and organs including lungs, liver, kidneys, spleen, intestine, as well as cardiac and skeletal muscle [24,26,29,31,51,55,60]. The pathology is comprised of severe pulmonary edema and congestion, multifocal hemorrhage, splenic atrophy, myocarditis, and necrosis [24,26,29,31,51,55,60]. In contrast, the animals infected via an aerosol lack extensive visceral pathology. This strongly suggests that the route of infection substantially alters the virus dissemination as well as the subsequent associated pathology [14–17].

EEEV localizes in the CNS of many mammalian species including humans regardless of the route of infection [11,12,14–33,35–50,52–69] [70]. Virus is readily detected in basal ganglia, hippocampus, frontal cortex, pons, thalamus, substantia nigra, mesencephalon, medulla

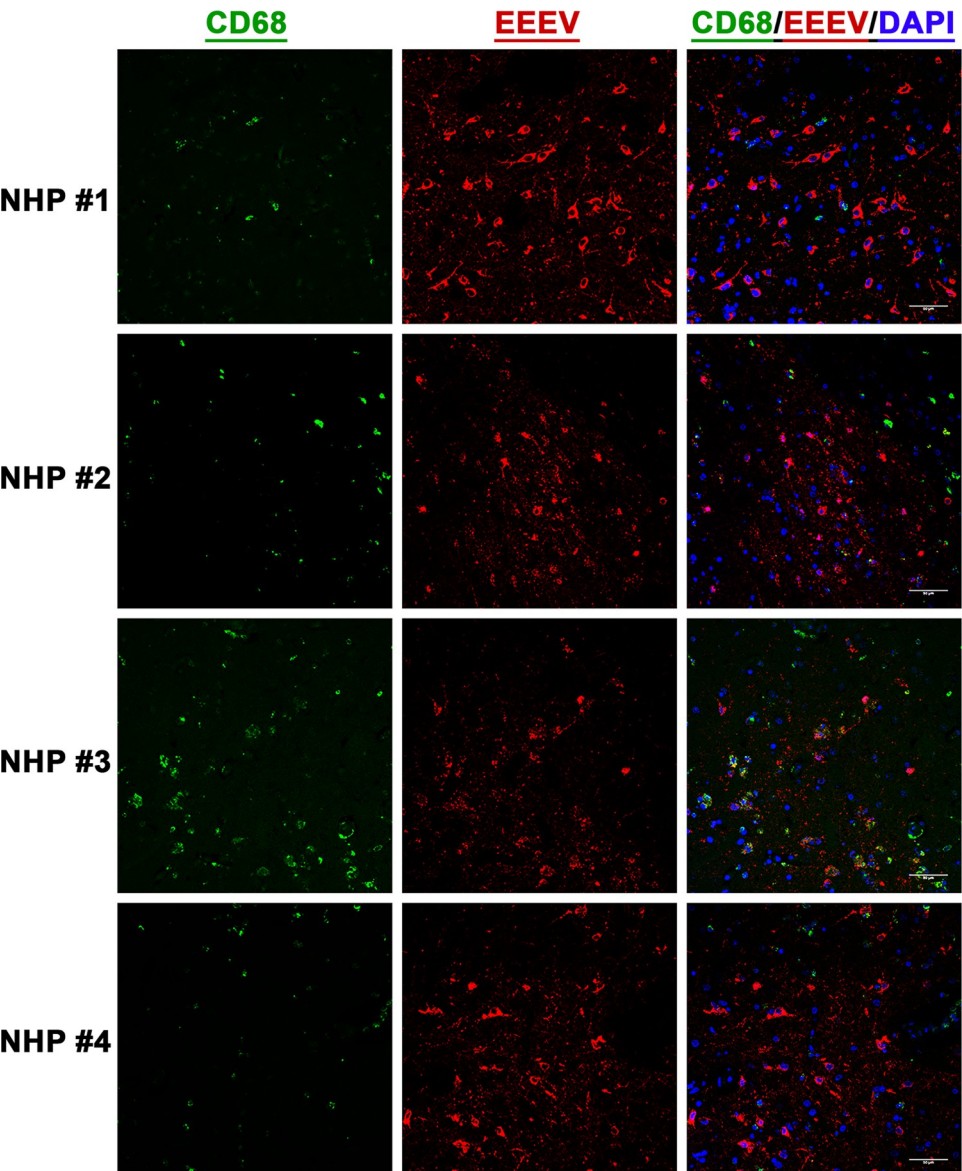

**Fig 11. The presence of EEEV RNA in the microglia of infected cynomolgus macaques.** Sections from the thalamus of each NHP were visualized via immunofluorescence assay. Sections were stained for CD68 (green), EEEV (red), and DAPI (blue).

oblongata, cerebellum, and spinal cord with minimal to moderate lesions [11,12,14–33,35–50,52–69]. These microscopic findings consist of neuronal degeneration and necrosis, neuropil vacuolation, gliosis, and satellitosis, neuronophagia, lymphocytic perivascular cuffing, lymphocytic meningitis, perivascular cuffs, neutrophil infiltrate, and microhemorrhage. The tropism of EEEV is predominantly limited to the neurons, however, astrocytes and microglia cells are also infected. The results of our study are in agreement with the majority of previously reported findings, however, there are several important differences. First, in our study, the majority of the cellular architecture in all brain regions remained intact and the focal degenerative and necrotic lesions were limited to the amygdala, hippocampus, corpus striatum, thalamus, mesencephalon, and medulla oblongata. Second, inflammatory lesions were limited and

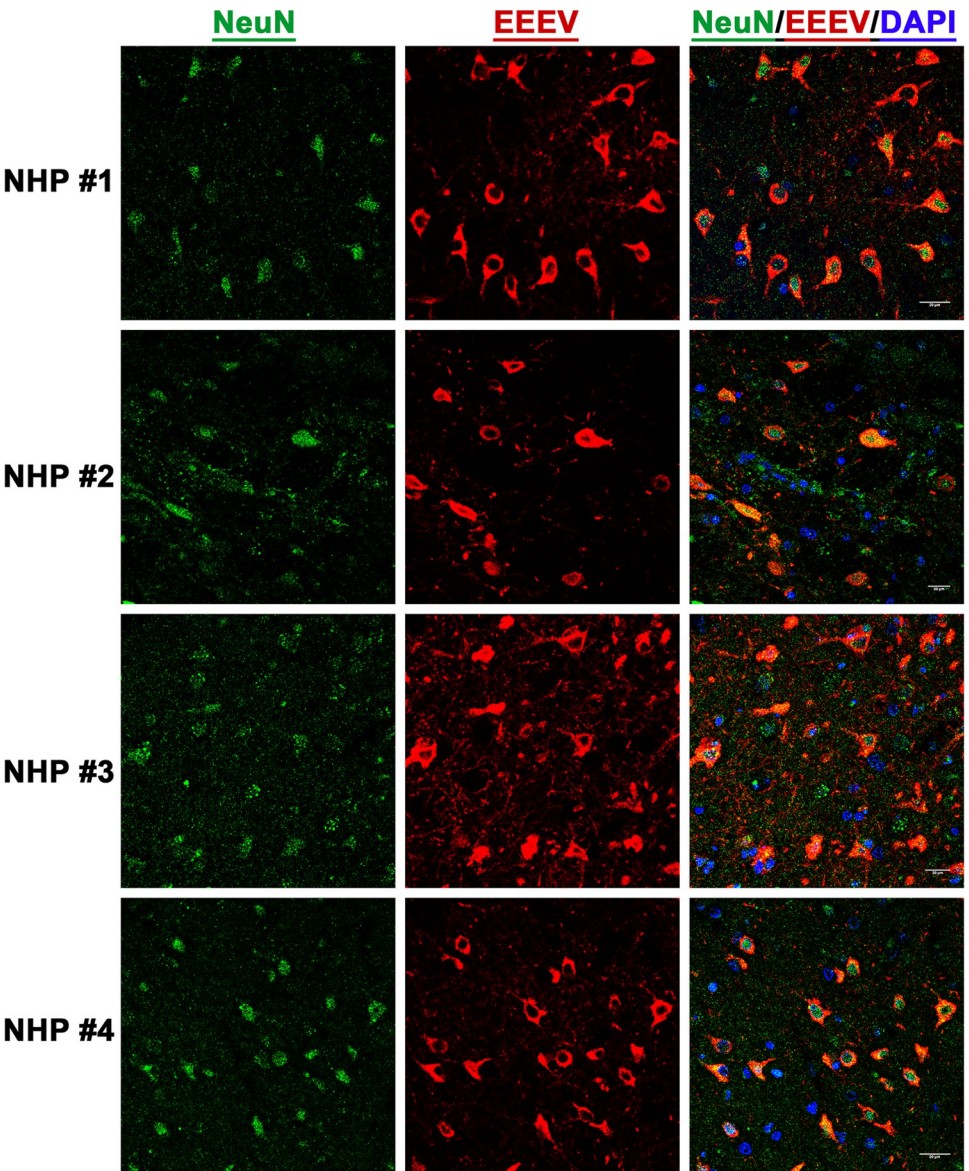

**Fig 12. The presence of EEEV RNA in the neurons of infected cynomolgus macaques.** Sections from the thalamus of each NHP were visualized via immunofluorescence assay. Sections were stained for NeuN (green), EEEV (red), and DAPI (blue).

comprised mainly of neutrophils. Third, the tropism of EEEV was almost exclusively to neurons. Fourth, microscopic findings were either absent or minimal in all sections of the spinal cord. These differences further highlight the alteration of virus pathogenesis following aerosol infection.

Limited studies have examined EEEV pathogenesis in the brain utilizing TEM [19,32,39,71]. These studies showed the presence of infectious particles, ~55–60 nm in diameter, localized almost exclusively in the extracellular spaces. The evidence of virus replication was either absent or rare in the tissues. Cytopathic vacuoles and nucleocapsid, ~28 nm in diameter, were observed in the cytoplasm of infected neurons and microglia. Infected and uninfected neurons, astrocytes, and microglia displayed dilated rough endoplasmic reticulum.

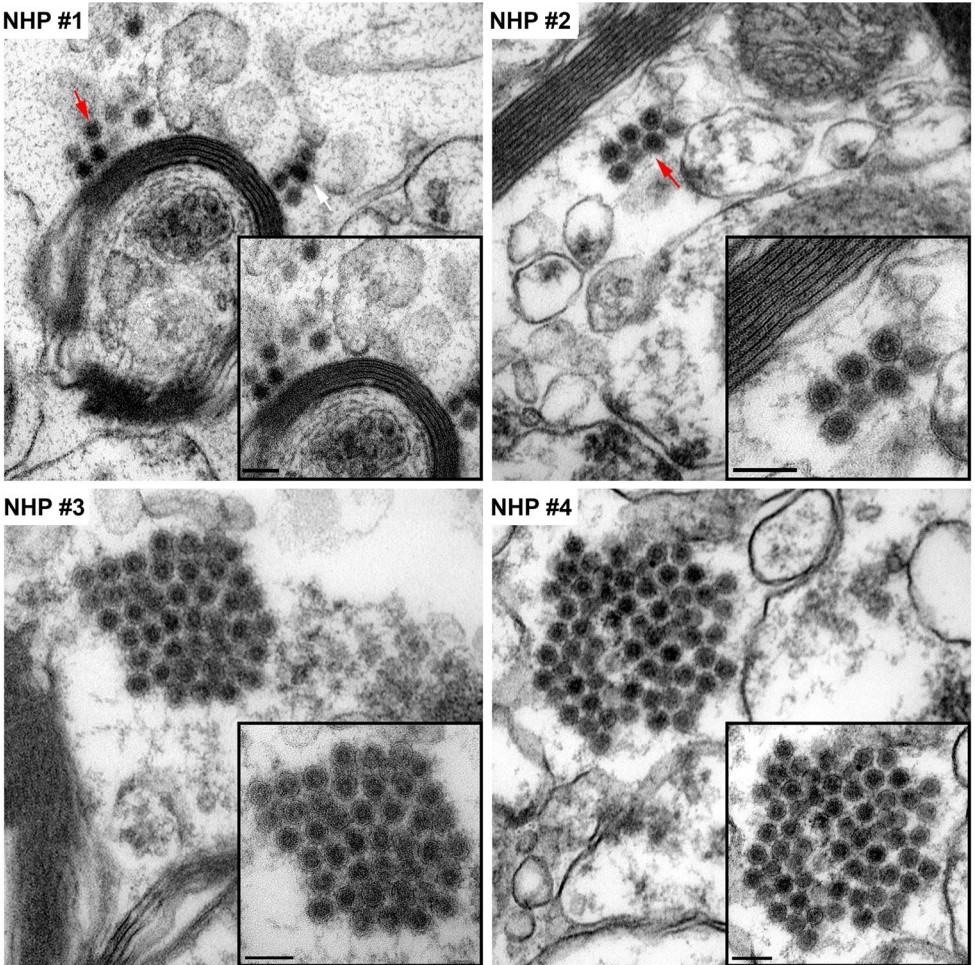

**Fig 13. The extracellular distribution of EEEV virions in the thalamus of infected cynomolgus macaques.** Sections from each NHP were examined and representative micrographs from each NHP are shown. Red arrows indicate virus particles. Bar = 200 nm.

Our study is in agreement with most of the previously reported findings, with one exception with regards to the size of virus particles and nucleocapsid. The infectious particle and nucleocapsid size were smaller in the previous human and mouse TEM studies than the recent cryo-electron microscopy (cryo-EM) studies that estimate the infectious particle and nucleocapsid size of ~65–70 and ~40–45 nm, respectively [3,72–74]. Our study is in agreement with the latter data. One potential explanation for this discrepancy is the shrinking effects of formalin fixation, dehydration, and paraffin embedding. The process of inactivation and embedding can reduce tissue size by up to 15% [75–78].

Axonal transport is an essential homeostatic process responsible for movement of RNA, proteins, and organelles within neurons [79]. Viruses including rabies, polio, West Nile, and Saint Louis encephalitis can utilize this critical mechanism and disseminate in the CNS via neuron-to-neuron spread [80–82]. The data from the present study showed that viral replication was limited to the olfactory bulb and proximal regions including the amygdala, hippocampus, thalamus, and hypothalamus. Viral RNA, proteins, and infectious particles were also detected in distal parts of the brain, however, minimal or no virus replication was detected. In addition, the infectious particles were present in the axon of neurons in all four NHPs. Thirty-

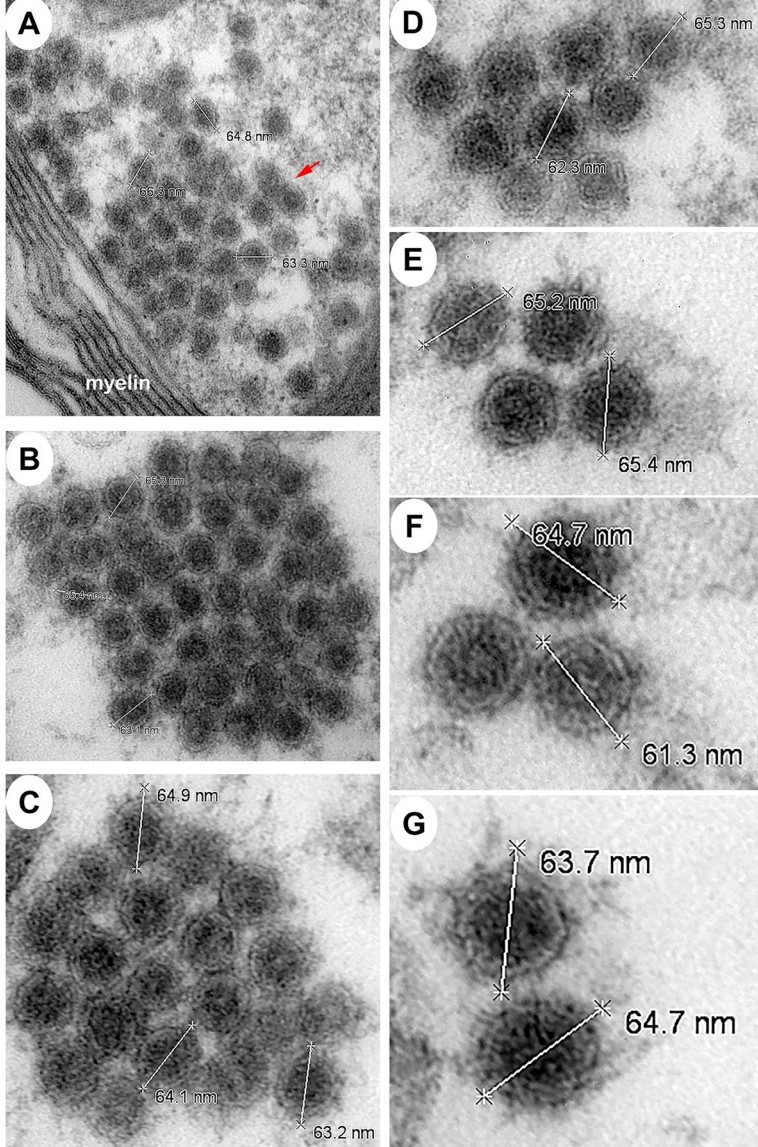

**Fig 14. The size of extracellular EEEV virions via transmission electron microscopy (TEM).** Sections from the thalamus of each NHP were examined and representative micrographs from NHPs are shown. Red arrow indicates virus particles.

five infectious particles were observed in a single 160 nm section of an axon. Taken together, these data strongly suggest that EEEV is able to rapidly spread throughout the CNS following aerosol challenge likely via axonal transport and warrants further investigation.

Many of the physiological parameters measured with advanced telemetry and 24-hr continuous monitoring were considerably altered following infection; temperature +3.0–4.2°C, respiration rate +56-128%, activity -15-76%, +5–22%, heart rate +67–190%, systolic blood pressure +44–67%, diastolic blood pressure +45–80%, ECG abnormalities, reduction in food/fluid intake and sleep, and EEG waves -99-+6,800%. Many of these parameters are under the control of the autonomic nervous system (ANS). The master regulator of the ANS is the hypothalamus which is comprised of numerous important nuclei that regulate these parameters; preoptic

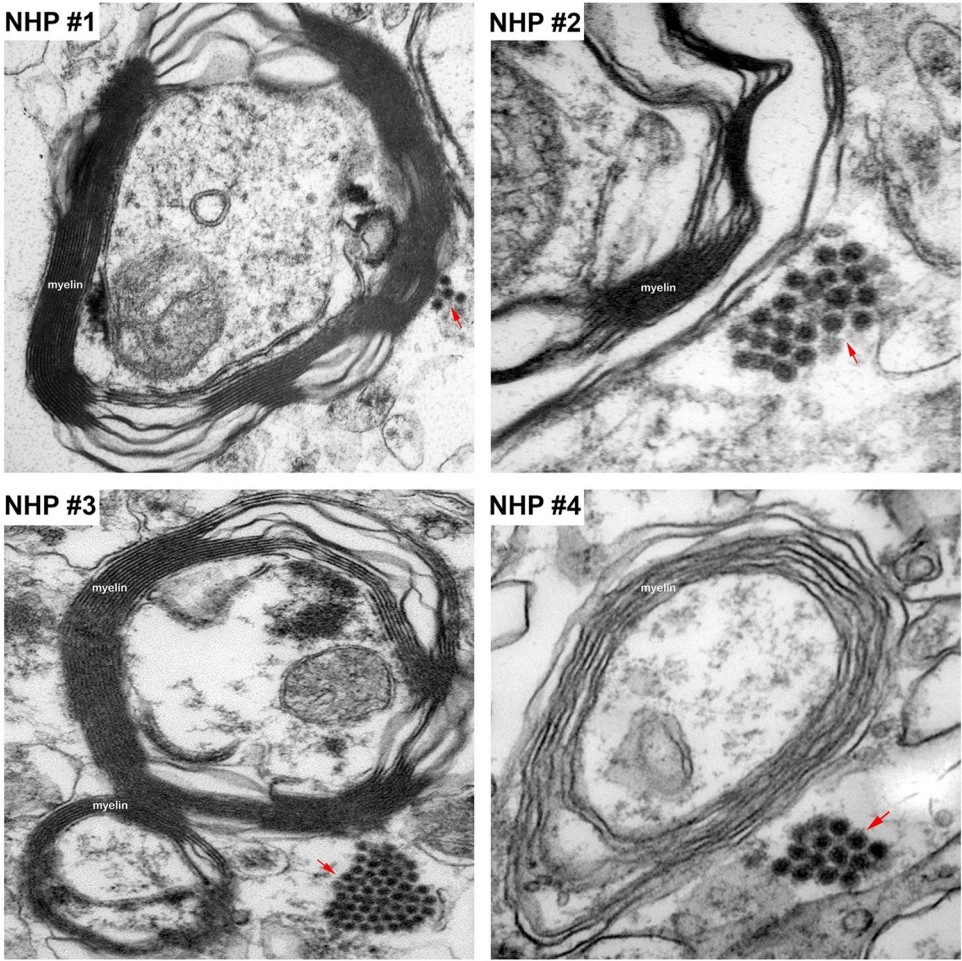

**Fig 15. The localization of EEEV virions around the myelin sheath of neurons via transmission electron microscopy (TEM).** Sections from the thalamus of each NHP were examined and representative micrographs from each NHP are shown. Red arrows indicate virus particles.

area (temperature), suprachiasmatic nuclei (circadian rhythm), paraventricular nuclei and supraoptic nucleus (hunger/satiety), tuberomamillary nucleus and the perifornical lateral (sleep), arcuate nucleus and paraventricular nucleus (blood pressure), arcuate nucleus (cardiac electrical system and heart rate), paraventricular nucleus, perifornical area, and dorsomedial hypothalamus (respiration) [83–90]. The hypothalamic nuclei are interconnected with many other regulatory centers such as the thalamus, basal ganglia, medulla oblongata, and others to exert control on important physiological parameters. The histopathology, ISH, IHC, and TEM data from this study shows the presence of viral RNA, proteins, and replication centers in the ANS control centers. These data suggest that EEEV infection in the brain likely produces disruption and/or dysregulation of the ANS control centers to produce rapid and extreme alterations in physiology and behavior to cause severe disease.

In many regions of the brain, EEEV infection produced minimal necrosis and inflammatory infiltrates, and majority of the cellular architecture remained intact. The observed necrosis and/or inflammation can contribute to severe disease, however, it cannot alone explain the fatal disease in the NHPs. One potential explanation of these results is that EEEV pathogenesis, in part, may be due to rapid local and global neuronal dysfunction. This hypothesis has been

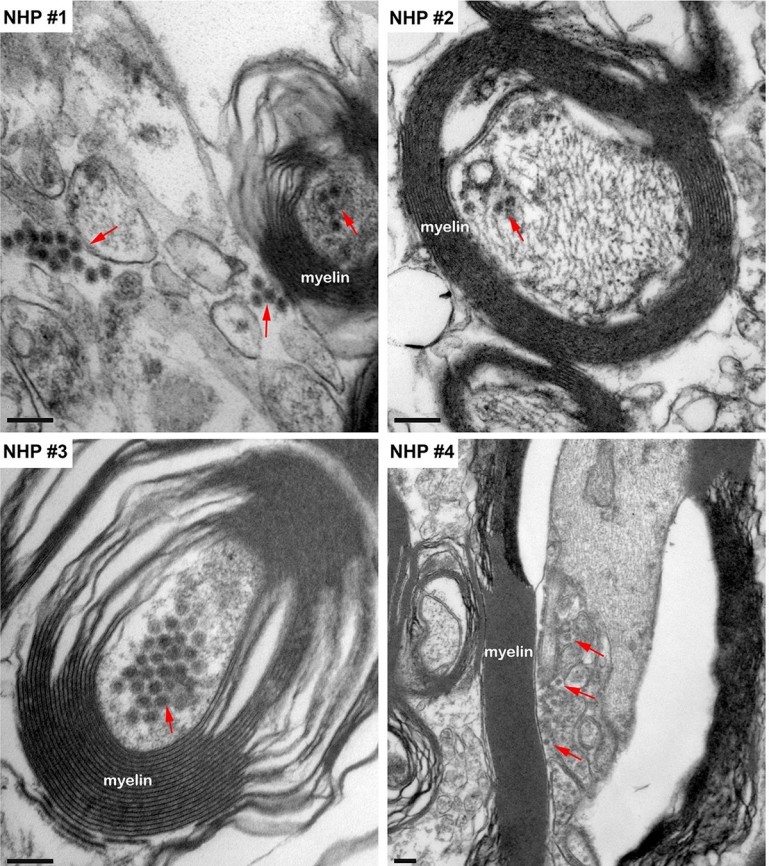

**Fig 16. The localization of EEEV virions within the axons of neurons via transmission electron microscopy (TEM).** Sections from the thalamus of each NHP were examined and representative micrographs of each NHP are shown. Red arrows highlight virus particles. Bar = 200 nm.

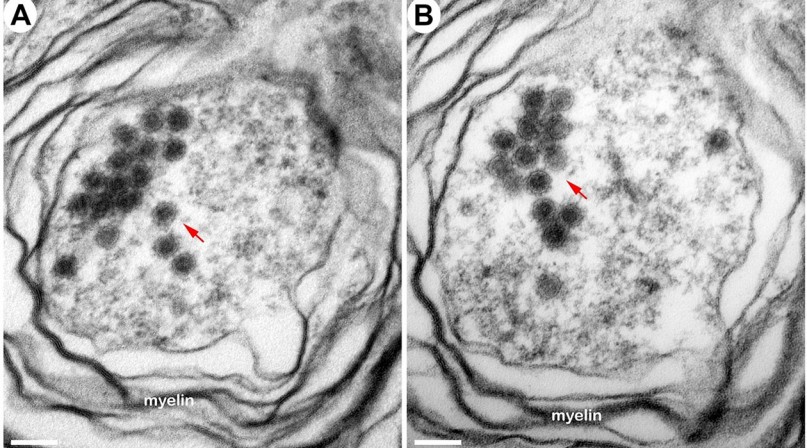

**Fig 17. The localization of EEEV virions within an axon of a neuron via transmission electron microscopy (TEM).** Sections from the thalamus of NHP #3 were examined. Two sequential sections, ~80 nm apart, of an axon are shown. Red arrows indicate virus particles. Scale bar = 100 nm.

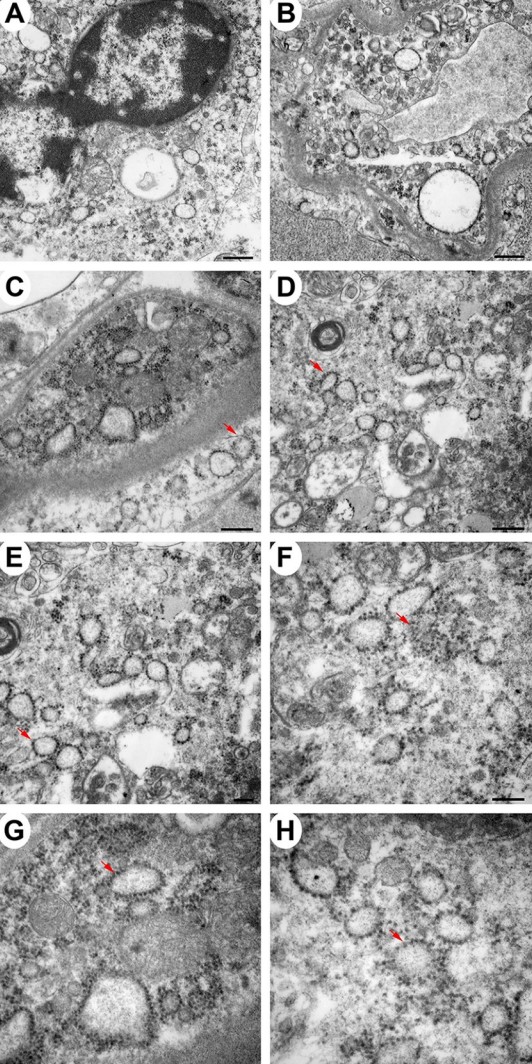

**Fig 18. The detection of cytopathic vacuoles in the cytoplasm of EEEV infected cells via transmission electron microscopy (TEM).** Sections from the thalamus of infected NHPs were examined. Micrographs of NHP #4 are shown. Scale bars: Panels A and B = 600 nm, C and D = 400 nm, E, F and G = 200 nm, and H = 100 nm.

investigated for a prototypic encephalitic virus, rabies virus (RABV). RABV can exert neuronal dysfunction by multiple mechanisms. RABV infection in neurons can induce degeneration of axons and dendrites without inflammation or cell death, axonal swelling, generation of toxic metabolites such as reactive oxygen species, decreased expression of housekeeping genes, impairment of both the release and binding of serotonin, and reduction in expression of voltage-dependent sodium channels [91–98]. Similar to RABV, the axonal transport of EEEV may also disrupt the transport of RNA, proteins, and/or organelles to produce neuronal dysfunction and leading to fatal outcomes. This hypothesis requires further investigation to elucidate the potential mechanism/s.

One of the NHP experienced a critical cardiovascular event and was subsequently euthanized. The investigation of the cardiac tissue showed no evidence of viral induced pathology, RNA, proteins, or host inflammatory response. In contrast, the brain tissues displayed some microscopic lesions as well as considerable presence of viral RNA and proteins, particularly in

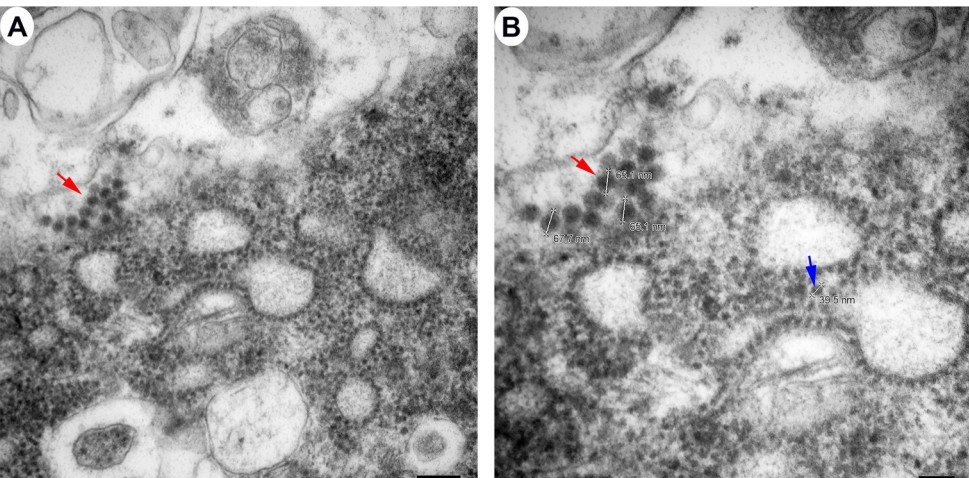

**Fig 19. The detection of cytopathic vacuoles, nucleocapsid, and budding virions in EEEV infected cells via transmission electron microscopy (TEM).** Blue and red arrows indicate nucleocapsid and virus particles, respectively. Sections from the thalamus of infected NHP #3 were examined. Scale bars: A = 200 nm, B = 100 nm.

the hypothalamus and medulla oblongata. These data suggest EEEV infection of the ANS control centers may have led to the dysregulation and/or disruption of the heart's electrical activity leading to a critical cardiac event. Lastly, the electrolyte imbalance due to considerable decrease in food/fluid intake in the NHP prior to the cardiac event may also contribute to the disruption and dysfunction the heart's electrical activity.

There are several important implications of our EEEV study regarding countermeasure development. First, the exposure by the aerosol route produces a rapid and profound infection of the CNS including the ANS control centers. Second, the axonal transport likely facilitates substantial neuron-to-neuron spread of virus. Third, the rapid viral spread in the CNS leads to considerable alterations of critical physiological parameters as early as ~12–36 hpi suggesting that the post-aerosol challenge window for therapeutic intervention may be short in the NHP model. Fourth, the presence of infectious virus within axons and the subsequent potential spread via axonal transport demonstrate the necessity for targeting small molecule or antibody therapeutics inside the axons to prevent/reduce infection and transport. Fifth, the investigation of therapeutics and vaccines in an aerosol NHP model should include monitoring of brain waves and comprehensive brain pathology following challenge.

In summary, EEEV initially replicated at olfactory bulb and was rapidly transported to distal parts of the brain likely utilizing axonal transport to facilitate neuron-to-neuron spread. Once within the CNS, the virus gained access to ANS control centers to cause the disruption and/or dysregulation of critical physiological parameters leading to NHPs meeting the euthanasia criteria ~106–140 hpi. The lack of diffuse necrosis in the CNS and ANS strongly suggests that EEEV pathogenesis is in part due to neuronal dysfunction. Lastly, the rapid spread of EEEV in the CNS and the subsequent substantial alteration of the critical physiological parameters in NHPs infected via aerosol route has important implications for countermeasure efficacy.

## Disclosure statement

The views expressed in this article are those of the authors and do not reflect the official policy or position of the U.S. Department of Defense, or the Department of the Army.

## Supporting information

**S1 Fig. The absence of EEEV RNA in visceral organs of infected cynomolgus macaques.** The tissues were collected at the time of euthanasia. The presence of viral RNA was visualized via *in situ* hybridization (ISH). ISH was performed on the tissues of all four NHPs. Bar = 200 um.
(TIF)

**S2 Fig. The absence of EEEV proteins in visceral organs of infected cynomolgus macaques.** The tissues were collected at the time of euthanasia. The presence of viral proteins was visualized via immunohistochemistry (IHC). IHC was performed on the tissues of all four NHPs. Bar = 200 um.
(TIF)

**S3 Fig. The extracellular distribution of EEEV virions in the thalamus of infected cynomolgus macaques.** Sections from NHPs were examined via transmission electron microscopy (TEM). Representative micrographs from each NHP are shown.
(TIF)

**S4 Fig. The localization of EEEV virions near synapses via transmission electron microscopy (TEM).** Sections from the thalamus of each NHP were examined and representative micrographs from each NHP are shown. NHP #1 (A), NHP #2 (B), NHP #3 (C), and NHP #4 (D). Blue and red arrows show synapses and infectious virus particles, respectively. Bar = 600 nm.
(TIF)

**S5 Fig. Transmission electron microscopy (TEM) micrographs of viral replication centers within the brain of non-human primates.** Top panels are representative electron micrographs of viral replication center (red asterisk) visible within the thalamus (A, E), amygdala (B, F), hippocampus (C, G) and hypothalamus (D, H) of a female non-human primate. The lower panels are also representative micrographs of the replication center in a male non-human primate. The number, size and intracellular localization of the replication center varies. A and F scale bar = 500 nm. B-E, G and H scale bar = 1 um.
(TIF)

**S6 Fig. The detection of EEEV particles enclosed within vesicles via transmission electron microscopy (TEM).** Sections from the thalamus of infected NHPs were examined and representative micrographs are shown. Red arrows indicate virus particles. Scale bar = 100 nm.
(TIF)

**S7 Fig. The detection of necrotic lesions in the thalamus of NHP #1 via transmission electron microscopy (TEM).** Red arrows indicate virus particles. Scale bars: A = 400 nm, B = 200 nm, C = 100 nm.
(TIF)

**S1 Table. List of tissue sections from each organ.**
(TIF)

## Author Contributions

**Conceptualization:** Margaret L. Pitt, Farooq Nasar.

**Formal analysis:** Janice A. Williams, Simon Y. Long, Xiankun Zeng, John C. Trefry, Sharon Daye, Paul R. Facemire, Patrick L. Iversen, Sina Bavari, Margaret L. Pitt, Farooq Nasar.

**Funding acquisition:** Margaret L. Pitt, Farooq Nasar.

**Investigation:** Janice A. Williams, Simon Y. Long, Xiankun Zeng, John C. Trefry, Sharon Daye, Sina Bavari, Margaret L. Pitt, Farooq Nasar.

**Methodology:** Janice A. Williams, Simon Y. Long, Xiankun Zeng, Kathleen Kuehl, April M. Babka, Neil M. Davis, Jun Liu, Sharon Daye.

**Project administration:** Sina Bavari, Margaret L. Pitt, Farooq Nasar.

**Supervision:** Sina Bavari, Margaret L. Pitt, Farooq Nasar.

**Writing – original draft:** Janice A. Williams, Simon Y. Long, Farooq Nasar.

**Writing – review & editing:** Janice A. Williams, Simon Y. Long, Xiankun Zeng, Kathleen Kuehl, April M. Babka, Neil M. Davis, Jun Liu, John C. Trefry, Sharon Daye, Paul R. Facemire, Patrick L. Iversen, Sina Bavari, Margaret L. Pitt, Farooq Nasar.

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
