## [Decision Letter · Decision Letter 0]

15 Jan 2021

Dear Dr Nasar,

Thank you very much for submitting your manuscript "Eastern Equine Encephalitis Virus Rapidly Infects and Disseminates in the Brain and Spinal Cord of Infected Cynomolgus Macaques Following Aerosol Challenge" for consideration at PLOS Neglected Tropical Diseases. As with all papers reviewed by the journal, your manuscript was reviewed by members of the editorial board and by several independent reviewers. In light of the reviews (below this email), we would like to invite the resubmission of a significantly-revised version that takes into account the reviewers' comments. 

We cannot make any decision about publication until we have seen the revised manuscript and your response to the reviewers' comments. Your revised manuscript is also likely to be sent to reviewers for further evaluation.

Sincerely,

Doug E Brackney, PhD

Associate Editor

Rebecca Rico-Hesse

Deputy Editor

Reviewer's Responses to Questions

**Key Review Criteria Required for Acceptance?**

**Methods**

-Are the objectives of the study clearly articulated with a clear testable hypothesis stated?

-Is the study design appropriate to address the stated objectives?

-Is the population clearly described and appropriate for the hypothesis being tested?

-Is the sample size sufficient to ensure adequate power to address the hypothesis being tested?

-Were correct statistical analysis used to support conclusions?

-Are there concerns about ethical or regulatory requirements being met?

Reviewer #1: No the methodology is not clearly articulated. The authors mention that all details can be found in a companion manuscript that is not included here. 

Line 393: the details of the virus stock should be described here as well. 

Line 405: the study design needs to be described here as well. 

Line 409: what exactly does the “aerosol route” mean? Was this large or small particle aerosols? Via a nebulizer? intrabronchial instillation? Why was the specific inoculum dose chosen?

Histopathology figures: all histopathology figures would benefit by the inclusion of symbols or arrowheads highlighting key pathological findings. This is critically important for the nonspecialist.

Reviewer #2: The study aims to provide descriptive pathological context to EEEV aerosol challenged cynomolgus macaques and is appropriately designed/powered for the analyses performed. Viral localization and cellular tropism are explored in four replicate study animals. As a descriptive study, statistical analyses were not performed. All appropriate ethical considerations have been met.

Comments

Were other methodologies considered for detecting viral replication such as staining for double-stranded RNA as a proxy for replication centers?

Reviewer #3: see summary and general comments below

**Results**

-Does the analysis presented match the analysis plan?

-Are the results clearly and completely presented?

-Are the figures (Tables, Images) of sufficient quality for clarity?

Reviewer #1: Line 123: the experimental results section should include some contextualization of the experimental parameters, e.g., dose, timing, n=?, etc. That is, what exactly was done to the animals to infect them. In addition, how did you confirm productive infection? Were there daily blood draws? Does aerosol inoculation result in systemic infection? Seroconversion?

Line 125: what was the time of euthansia?

Line 133: what companion manuscript?

Line 244-245: are there differences in the way virus is administered in the different animal models that could explain differences in pathology and replication?

Line 226-269: it isn’t clear if any of these referenced studies were done with macaques. 

Line 319-324: None of the telemetry data are presented in the results.

Line 381: this is the first mention of the timing of clinical signs after inoculation. This should be more thoroughly described in the results section.

Reviewer #2: Results are well organized and figures are of sufficient quality to support the findings.

Comments

Figures (General): Since sections from control animals are unavailable for comparison, please provide callouts (arrows or otherwise) to example histology lesions described in text for additional clarity.

Reviewer #3: see summary and general comments below

**Conclusions**

-Are the conclusions supported by the data presented?

-Are the limitations of analysis clearly described?

-Do the authors discuss how these data can be helpful to advance our understanding of the topic under study?

-Is public health relevance addressed?

Reviewer #1: Discussion: overall, the discussion is comprehensive but it in many places it borders on results. In addition, it was unclear in many instances whether results from the companion manuscript were simply being described again here. I suggest trying to tighten the discussion to only include the information necessary to understand the study described here.

Reviewer #2: The authors’ conclusions are supported by the pathological findings. Mechanisms of virus spread within neural tissue is extrapolated from virus localization at the tissue (IHC/ISH) and cellular level (TEM). The authors adequately contextualized their findings with previous studies in NHPs and small animal models of EEEV infection, including aerosol exposure. 

Comments

Lines 278-279: Add context to expected cellular targets for infection (i.e. inhibition in myeloid cells described in Trobaugh et al. 2014 and others)

Reviewer #3: see summary and general comments below

**Editorial and Data Presentation Modifications?**

Reviewer #1: Grammar and syntax could be improved throughout and the manuscript would benefit from a careful editorial review.

Reviewer #2: Were lymphatics (other than spleen) also examined for EEEV (RNA or antigen)? 

Lines 393-394: If a citation to the companion manuscript is unavailable at the time of acceptance, please provide sufficient detail about virus stock, including passage history.

Lines 439-441: Please provide a citation or additional information for non-commercial reagent. What alphavirus antigen(s) was/were used to derive polyclonal antibody?

Lines 454-458: Please provide a citation or additional information for non-commercial reagent. Was Rabbit anti-EEEV raised against whole virus or specific antigenic regions?

Reviewer #3: see summary and general comments below

**Summary and General Comments**

Reviewer #1: Here the authors have produced a paper investigating pathogenic outcomes in crab-eating macaques that were infected with eastern equine encephalitis virus. Using aerosol inoculation, they demonstrate that the virus is found in many parts of the brain and spinal cord, and the virus gets there via the olfactory bulb. While these results are interesting and important, conclusions from this work are diminished by the lack of information presented pertaining to overall infection parameters. In addition, the authors reference a companion manuscript that contains additional details but some of these details are critical for interpretation of the work presented herein. In general, not enough information is presented on the overall study design and acute infection parameters. The work seems to be divided in the two papers strictly along the lines of pre vs. post euthanasia analyses—althought this is difficult to assess because the companion manuscript was not included. You basically need to switch back and forth between the two to manuscripts to understand what is happening, and that isn’t possible with submission of this manuscript. I am not clear on the justification for splitting the study into separate manuscripts, so unless this is clearly delineated I would strongly encourage the team to submit a single combined manuscript. At a minimum this manuscript needs to be written in a manner that it is stand alone and can be fully understood without significant references of the companion manuscript.

Reviewer #2: Williams et al. infected cynomolgus macaques with eastern equine encephalitis virus (EEEV) via aerosol challenge and explored the clinical pathology and cellular tropism in detail at the time of necropsy. The authors used a collection of techniques to identify viral RNA and antigens in host organs and describe clinically significant lesions associated with disease. This manuscript is a companion to additional work exploring the virology and clinical course of disease. The authors present the first comprehensive analysis of clinical pathology associated with EEEV aerosol exposure in cynomolgus macaques, building upon the work of Reed et al. 2007 and others. Furthermore, this work explores in detail the cellular tropism and pathology associated with EEEV replication in the brain.

Reviewer #3: Williams et al describe a pathogenesis study in cynomolgus macaques exposed by aerosol to eastern equine encephalitis virus (EEEV). This appears to be part of a larger study where some of the results are described in a companion manuscript that focuses on the response to infection measured by telemetry. The current manuscript describes the microscopic lesions in the tissues collected when the animals were euthanized on days 4-5 post-infection. Major findings include virus was only detected in the brain and spinal cord, but limited to no histopathological lesions were observed. Their results suggest that EEEV initially replicates in the olfactory bulb and is then transported throughout the brain by the neuronal axonal transport system assisting neuron-to-neuron spread. Neurons were found to be the primary target, but their data suggests that this leads to neuronal dysfunction rather than death. While this study is largely descriptive, the in-depth characterization of the pathogenesis in the tissues (particularly by EM) is an important contribution to the field. However, the manuscript would be significantly improved by adding clarification for some of their conclusions and consolidating/reducing the number of figures to the major findings. 

Major comments:

1. It is difficult to follow some aspects of the results because the authors keep referring to the “companion manuscript” but no reference is provided. More details need to be provided. At a minimum the reference needs to be provided. It seems like these two manuscripts should be published together. 

2. The number of figures needs to be reduced and should be consolidated to present/highlight major findings. For example, Figure 1 is not necessary because it shows no histopathology. For other figures, the authors should show representative images and not the image from each animal to reduce the number of figures. For example, figures 3, 4, 5, and 6 could all be combined into one figure with representative images. The same for figures 7, 8, 9. Figures 10-12 could be combined into one as well.

3. Some of the major findings and statements conflict with previous reports on the pathogenesis of EEEV in NHPs and are not discussed. For example, line 244-245 states that the dissemination of EEEV following aerosol infection has not been investigated in previous macaque studies, but this is not accurate. For example, Roy et al. 2013 found high viral loads in peripheral tissues of animals that succumbed to infection on days 4-6 post-infection. Additionally, the pathology data for the Reed et al. 2007 JID article is published by Steele and Twenhafel 2010 Vet. Pathol. 47, 790—805. This article describes the key pathological features in the brain as severe meningoencephalomyelitis. These results seem to conflict with the current study where the authors’ state that “little or no pathological lesions” were observed. Also, pathogenesis studies of EEEV in NHPs by subcutaneous exposure (to mimic mosquito transmission) have been completed and should be discussed.

4. The authors need to make clearer that their findings are specifically for animals who have been exposed to EEEV by aerosol. For example, this could be clarified in lines 227-228 and 242-243. It would be beneficial if the authors would expand their discussion about what is known about the pathogenesis of EEEV primarily in humans (not lump all mammalian species as a whole; lines 257-258) and how their results compare. For example, vasculitis is an important pathological feature in humans infected with EEEV and have been reported in other aerosol exposure studies of EEEV in cynomolgus macaques, but I do not see it discussed in the current study. Why does it seem that the current study has different conclusions/findings compared to other aerosol exposure studies in cynomolgus macaques that did observe significant pathology in the brain? 

Minor comments:

Line 135: Shouldn’t this be reported as PFU/g since it is the titer in tissue?

Line 139: Should “vacuolation of the neutrophil” be vacuolation of the neuropil?

All H&E images: Arrows should be added to highlight findings (i.e. necrosis, neutrophilic infiltrates, etc) for readers that are not used to looking at histopathology images.

PLOS authors have the option to publish the peer review history of their article (what does this mean?). If published, this will include your full peer review and any attached files.

Reviewer #1: No

Reviewer #2: No

Reviewer #3: No
---

## [Decision Letter · Decision Letter 1]

5 Oct 2021

Dear Dr Nasar,

Thank you very much for submitting your manuscript "Eastern Equine Encephalitis Virus Rapidly Infects and Disseminates in the Brain and Spinal Cord of Cynomolgus Macaques Following Aerosol Challenge" for consideration at PLOS Neglected Tropical Diseases. As with all papers reviewed by the journal, your manuscript was reviewed by members of the editorial board and by several independent reviewers. The reviewers appreciated the attention to an important topic. Based on the reviews, we are likely to accept this manuscript for publication, providing that you modify the manuscript according to the review recommendations. 

Generally speaking the reviewers were happy with the changes that were made to the manuscript; however, two reviewers felt that there were a number of issues that needed to be further discussed in detail within the discussion section of the manuscript.

Sincerely,

Doug E Brackney, PhD

Associate Editor

Rebecca Rico-Hesse

Deputy Editor

Generally speaking the reviewers were happy with the changes that were made to the manuscript; however, two reviewers felt that there were a number of issues that needed to be further discussed in detail within the discussion section of the manuscript.

Reviewer's Responses to Questions

**Key Review Criteria Required for Acceptance?**

**Methods**

-Are the objectives of the study clearly articulated with a clear testable hypothesis stated?

-Is the study design appropriate to address the stated objectives?

-Is the population clearly described and appropriate for the hypothesis being tested?

-Is the sample size sufficient to ensure adequate power to address the hypothesis being tested?

-Were correct statistical analysis used to support conclusions?

-Are there concerns about ethical or regulatory requirements being met?

Reviewer #1: (No Response)

Reviewer #4: The objectives of the study are articulated but they should explain why the effort is limited to aerosol exposure only.

This reviewer questions if a detailed analysis of pathology from aerosol exposure with EEE gives a thorough picture of EEE pathology in NHPs.

What is the power calculation for the sample size [number of macaques (4)] used in the study to analyze data significance? How was this number determined? I realize NHP studies are costly.

yes, appears ethical and regulatory requirements have been met.

Reviewer #5: Methods of the manuscript are acceptable; however, alternative approaches that might improve the capacity for detection of replicating virus such as in situ hybridization for negative strand RNA would have been useful rather than reporting cytopathic vacuoles. Branched chain amplification coupled with RNAscope should be extremely sensitive for detection of replication intermediates.

Directly relevant to the hypothesized mechanism of disease progression (neuronal dysfunction) have the investigators examined means for characterizing this with specifically? Quantifying possible axonomal degeneration or some other marker for neuronal function?

**Results**

-Does the analysis presented match the analysis plan?

-Are the results clearly and completely presented?

-Are the figures (Tables, Images) of sufficient quality for clarity?

Reviewer #1: (No Response)

Reviewer #4: Otherwise, this work provides pathogenesis details not addressed in previous EEEV challenges following aerosol exposures. Having 21 figures and 5 supplemental figures makes for a rather (figure) dense paper and this reviewer considers it important where possible to consolidate data presentation critical for this paper or at least move some of the paper figures to the supplemental section.

Reviewer #5: Results presented are adequate but I would have liked to have seen some specific data regarding infiltrates and a rigorous assessment of whether EEEV RNA was identified. This addresses the central hypothesis generated and could also be related to reduced inflammation due to a lack of peripheral replication due to the route of exposure.

**Conclusions**

-Are the conclusions supported by the data presented?

-Are the limitations of analysis clearly described?

-Do the authors discuss how these data can be helpful to advance our understanding of the topic under study?

-Is public health relevance addressed?

Reviewer #1: (No Response)

Reviewer #4: Others have already shown encephalitic alphaviruses can gain entry to the CNS via the olfactory tract following intranasal or aerosol routes of challenge with neurovirulent alphaviruses in animal models. However, the studies by Williams et al. show little (to no) such evidence of virus replication outside the CNS as well as in the brain. This is a curious finding. Encephalitic alphaviruses (WEEV and VEEV) definitely show virus replication in the brain of mice following intranasal and peripheral routes of challenge but the authors here show minimal evidence of viral replication in macaque CNS. This is addressed in the Discussion but only in general terms and may highlight viral pathogenesis differences using different animal models.

Reviewer #5: Necrotic lesions, although reported to be very rare in this study, could significantly contribute to disease progression. The authors should indicate this as a possibility within the Discussion section.

More specific discussion regarding this route of infection should be made for interpretation of the results presented herein with previous reports. The lack of peripheral replication could have a significant impact on inducing innate immune responses and subsequent inflammation within the CNS.

**Editorial and Data Presentation Modifications?**

Reviewer #1: (No Response)

Reviewer #4: No comment

Reviewer #5: The authors refer to a North American lineage virus. Functionally, EEEV is the only virus within this group after the South American lineage viruses were reclassified as Madriagas virus.

**Summary and General Comments**

Reviewer #1: No general comments, reviewers responded thoughtfully to reviewers concerns. 

The authors made a majority of the recommended changes requested during initial peer-review of this manuscript and if the changes were not made, a sufficient explanation was provided. The changes significantly enhanced the credibility and scientific nature of the manuscript. The readers of the article can now fully understand the scientific methods used throughout this study and accurately interpret the scientific findings without bias or incomplete information.

Reviewer #4: PTND-20-02124R1: This work by Williams et.al. sheds light on EEEV pathogenesis in the CNS of an NHP (macaques) following aerosol exposure of EEEV. The team used advanced telemetry to measure the animal’s physiological parameters following aerosol exposure. Others have already shown encephalitic alphaviruses can gain entry to the CNS via the olfactory tract following intranasal or aerosol routes of challenge with neurovirulent alphaviruses in animal models. However, the studies by Williams et al. show little (to no) such evidence of virus replication outside the CNS as well as in the brain. Encephalitic alphaviruses (WEEV and VEEV) definitely show virus replication in the brain of mice following intranasal and peripheral routes of challenge but the authors here show minimal evidence in macaques. This is addressed in the Discussion but only in general terms but may highlight pathogenesis differences using different animal models. Additionally, outside of possible exposures through bioterrorist events or laboratory accidents, does it make sense to report aerosol exposures only and pass on any attempts to define or describe outcomes of virus pathology/challenge from peripheral exposures (the current danger from mosquito-borne alphaviruses). I realize NHP studies are expensive but some of the claims reported here appear to contradict previous pathogenesis studies in NHP and other animal models and there is minimal discussion of this point. Are the authors assuming the pathologies will be the same for aerosol vs peripheral exposures in NHPs? Others have shown differences in virus distribution and pathology in the CNS when using peripheral vs intranasal virus challenges and the authors acknowledge this in the Discussion. Aerosol challenges may not give the complete story in terms of pathology and may be specific to route of virus entry.

Reviewer #5: (No Response)

PLOS authors have the option to publish the peer review history of their article (what does this mean?). If published, this will include your full peer review and any attached files.

Reviewer #1: No

Reviewer #4: No

Reviewer #5: No

Figure Files:

Data Requirements:

Reproducibility:

References

---

## [Editor Report · Decision Letter 2]

9 Dec 2021

Dear Dr Nasar,

We are pleased to inform you that your manuscript 'Eastern Equine Encephalitis Virus Rapidly Infects and Disseminates in the Brain and Spinal Cord of Cynomolgus Macaques Following Aerosol Challenge' has been provisionally accepted for publication in PLOS Neglected Tropical Diseases.

Best regards,

Doug E Brackney, PhD

Associate Editor

Rebecca Rico-Hesse

Deputy Editor

---

## [Editor Report · Acceptance letter]

10 Feb 2022

Dear Dr Nasar,

We are delighted to inform you that your manuscript, "Eastern Equine Encephalitis Virus Rapidly Infects and Disseminates in the Brain and Spinal Cord of Cynomolgus Macaques Following Aerosol Challenge," has been formally accepted for publication in PLOS Neglected Tropical Diseases.

Best regards,

Shaden Kamhawi

co-Editor-in-Chief

Paul Brindley

co-Editor-in-Chief
